# Interannual variation of a bottom cold water mass in the Seto Inland Sea, Japan

Junying Zhu[1], Jie Shi[2,3], Xinyu Guo[4]

[1]College of Marine Sciences, Hainan University, Haikou 570228, China

[2]Key Laboratory of Marine Environment and Ecology, Ocean University of China, Ministry of Education, 238 Songling Road, Qingdao 266100, China

[3]Laboratory for Marine Ecology and Environmental Sciences, Qingdao National Laboratory for Marine Science and Technology, Qingdao 266071, China

[4]Center for Marine Environmental Studies, Ehime University, 2-5 Bunkyo-Cho, Matsuyama 790-8577, Japan

*Correspondence to*: Xinyu Guo (guoxinyu@sci.ehime-u.ac.jp)

**Abstract.** A bottom cold water mass (BCWM) is a widespread physical oceanographic phenomenon in coastal seas, and its temperature variability has an important effect on the marine ecological environment. In this study, the interannual variation of the BCWM in Iyo-Nada (INCWM), a semi-enclosed bay in the Seto Inland Sea, Japan, from 1994 to 2015 and its response to air-sea heat flux change were investigated using monthly observational data and a hydrodynamic model.

Surrounded by the isotherm of 18 °C, the observed multi-year average water temperature inside the INCWM was 17.58 °C with a standard deviation of 0.27 °C, while the mean area of INCWM was $5.73 \times 10^5$ m$^2$ with a standard deviation of $4.35 \times 10^5$ m$^2$. The interannual variation of average water temperature of INCWM showed a negative correlation with its area that indicates a low temperature corresponds to a big area. In addition, the interannual variation of the average temperature inside INCWM showed positive correlations with the local water temperature from April to July and with remote water

temperature below 10 m in an adjacent strait in July. Differing from previously studied BCWMs, which had interannual variations depending closely on the water temperature before the warming season, the interannual variation of INCWM is more sensitive to the air-sea heat flux during the warming season than that in the previous winter. Further, by comparing several BCWMs, we found that the BCWM size is a key factor in understanding the heat transfer process responsible for the interannual variation of BCWMs in coastal seas. These findings will help us to understand the response of bottom cold water

mass in coastal seas to sea surface forcing change.

## 1 Introduction

With global climate change, interannual variations in water temperature in coastal oceans are attracting significant attention (Lin et al., 2005; Park et al., 2015; Chen et al., 2020). However, most studies consider sea surface temperature. The lack of long-term observations limits the studies on the bottom water temperature, even though its interannual variations are

important for understanding how coastal oceans respond to atmospheric changes (Simpson et al., 2011; Turner et al., 2017).

A bottom cold water mass (BCWM), also called "cold pool", is the water trapped on the bottom layer as a result of seasonal thermoclines during stratified seasons. It is characterized by lower temperatures than the surrounding waters and has been reported to occur in many shelf seas, such as the Yellow Sea (Wei et al., 2010), Irish Sea (Hill et al., 1994), Middle Atlantic Bight (Lentz, 2017), North Sea (Brown et al., 1999), Bering Sea (Zhang et al., 2012) and Seto Inland Sea (Yu and Guo, 2018). As BCWMs occur every year with an annual cycle of forming during the warming season (from spring to summer), being the strongest in summer, and disappearing in early fall, they are good indicators to demonstrate the response of the coastal sea to atmospheric change. In addition, interannual variations in BCWMs have been reported to affect marine ecosystems (Stabeno et al., 2012; Wang et al., 2014; Abe et al., 2015). Therefore, clarifying their interannual variations and controlling factors are helpful for understanding marine ecosystem changes.

Limited by the absence of long-term observations, only the Yellow Sea Cold Water Mass and the Middle Atlantic Bight Cold Pool have been studied for their interannual variation and causes (Yang et al., 2014; Coakley et al., 2016; Li et al., 2017; Zhu et al., 2018; Chen and Curchitser, 2020), both of which exhibited apparent interannual variations. Specifically, the interannual variations of the Yellow Sea Cold Water Mass are closely related to the air-sea heat flux in the previous winter (Wei et al., 2010; Park et al., 2011; Li et al., 2015; Zhu et al., 2018). After studying the Middle Atlantic Bight Cold Pool, Chen and Curchitser (2020) suggested that its temperature interannual variations during stratified seasons were controlled by both the previous winter temperature and abnormal warming/cooling due to total oceanic advection. They also pointed out that the winter (mid-January to March) temperature anomaly was the primary factor in determining the interannual variability of temperature anomaly near bottom cold pool region during the stratified seasons. Nevertheless, there remains little information regarding the interannual variations of BCWMs in other coastal seas.

The Seto Inland Sea is the largest semi-closed coastal sea in western Japan, with an average depth of 38 m and a surface area of 23,000 km$^2$ (Fig. 1a). It opens to the Pacific Ocean via Bungo Channel and Kii Channel and is divided into several shallow basins by narrow straits. The presence of a BCWM in summer has been confirmed in several of its wide basins, including the Iyo-Nada (Takeoka et al., 1993; Yu et al., 2016), Suo-Nada (Chang et al., 2009), Hiuchi-Nada (Guo et al., 2004), and Harima-Nada (Chang et al., 2009). In this study, we focus on the BCWM in Iyo-Nada, which connects the Bungo Channel with the Hayasui Strait (Fig. 1b). The depth of Hayasui Strait is more than 100 m and the water temperature is homogenous throughout the year because of strong tidal mixing (Kobayashi et al., 2006). From May to July, a density-induced gravitational circulation occurs as the bottom water flows from the Hayasui Strait to Iyo-Nada, whereas the surface water flows in the opposite direction (we draw this circulation in a schematic diagram in Fig. 6). This circulation can be enhanced in July by an abrupt increase in river discharge into the Seto Inland Sea (Yu et al., 2016; Yu and Guo, 2018).

Previous studies have examined the seasonal change in the BCWM in Iyo-Nada (hereafter, we use "INCWM" to denote the Iyo-Nada BCWM). With the development of stratification, the water temperature of the INCWM increases by 8-10 ℃ from early April to August. Yu and Guo (2018) suggested that seasonal warming in the INCWM results from both lateral heat advection from the surrounding water and vertical diffusion from the surface layer, in which lateral heat advection

contributes more than vertical diffusion. However, it is still unknown how the INCWM changes and how these lateral and

vertical processes affect the INCWM on an interannual scale.

The remainder of this study is organized as follows. In Section 2, we introduce the long-term observation data and a hydrodynamic model. In Section 3, we describe the interannual variation in the INCWM and its relationship with the surrounding water. In Section 4, we discuss the sensitivity of INCWM characteristics to air-sea heat flux changes by several sensitivity numerical experiments. Then we compare the INCWM with other BCWMs in coastal seas. Finally, we

summarize the main results in Section 5.

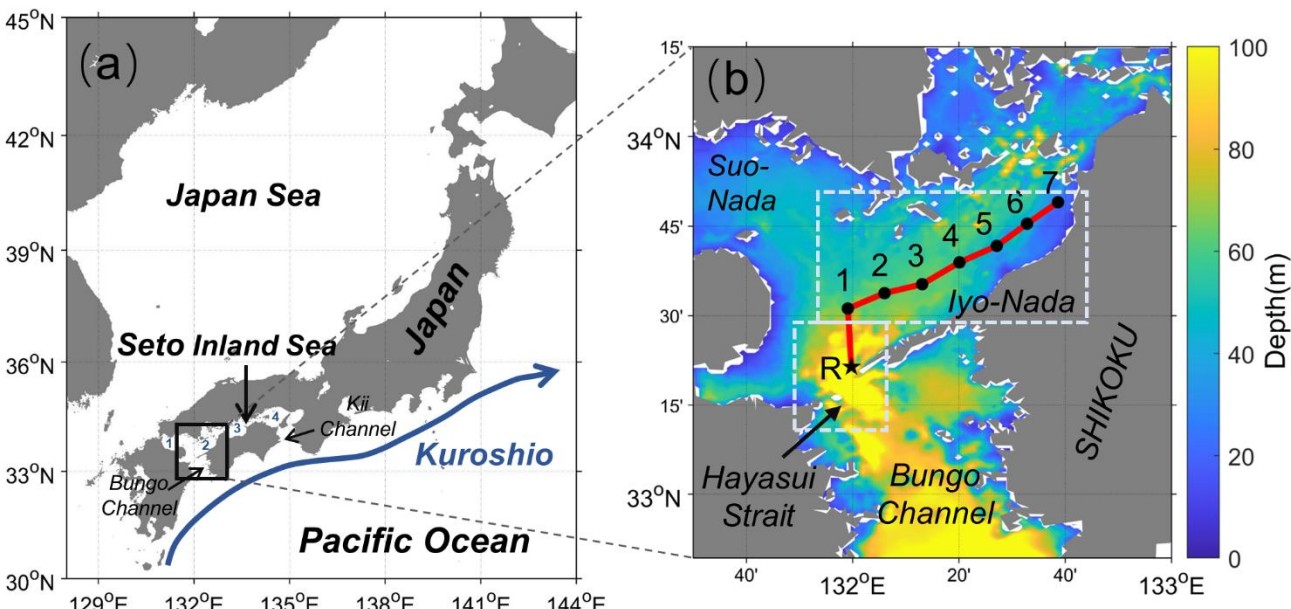

**Figure 1: Location of (a) Seto Inland Sea and (b) observation stations. The numbers from 1 to 4 in (a) represent Suo-Nada, Iyo-Nada, Hiuchi-Nada and Harima-Nada, respectively. Observation stations are represented by black dots and stars. The areas**

**enclosed in the dotted boxes in (b) represent Iyo-Nada (131.9-132.8°E, 33.45-33.75°N) and the Hayasui Strait (131.8-132.1°E, 33.15-33.45°N), which were used in the numerical sensitivity experiments.**

## 2 Methods

### 2.1 Long-term observation data

The Prefectural Fishery Research Centers around the Seto Inland Sea have conducted ship-based hydrographic observations

almost every month since 1971 to monitor water quality. For this study, we collected the observed water temperature data inside the Iyo-Nada (stations 1-7) and the Hayasui Strait (station R) during 1994-2015 from Ehime Prefectural Fishery

Research Centers. The locations of stations are shown in Fig. 1b. The water temperature was measured by ALEC CTD carried by survey vessel "Yoshuu". The measurement accuracy is 0.001 °C. Observation year for each month in Iyo-Nada and Hayasui Strait during 1994-2015 are shown in Table S1. For each month, there are at least 17 years of observations in Iyo-Nada. The observation data almost cover depths of 0 m, 10 m, 20 m, and 50 m at all stations as well as additional 75 m depth measurements at station R though the depth of observation varies slightly in different years. We interpolated water temperature to the deepest observed location at each station to show the vertical distribution of temperature.

## 2.2 Model configuration

Herein, we used a three-dimensional free-surface primitive equation ocean model to examine the factors influencing interannual variation in the INCWM. This model is based on the Princeton Ocean Model, and its configuration is described in detail by Zhu et al. (2019). Model domain is 130.98°E-135.5°E, 32.8°N-34.8°N covering the entire Seto Inland Sea. The model has a horizontal resolution of around 1 km (1/80° in the zonal direction and 1/120° in the meridional direction), and a vertical resolution of 21 sigma levels. The initial temperature and salinity fields in January were produced by merging the Marine Information Research Center dataset in the SIS region and the model results of Guo et al. (2003). The boundary conditions including de-tided current velocity, temperature, salinity, and surface elevation were obtained from the diagnostic model of Guo et al. (2004). Four major tidal constituents ($M_2$, $S_2$, $O_1$ and $K_1$) were considered and the daily river discharges averaged over 24 years (1993–2016) from the Ministry of Land, Infrastructure and Transport were used in the model. Multi-year averaged daily surface fluxes of momentum, heat and fresh water were used to drive model (Zhu et al., 2019). The daily wind stress was based on hourly averaged results of wind stress, which was calculated by wind velocity from the Grid Point Value of Meso-Scale Model (GPV-MSM) (http:// database.rish.kyoto-u.ac.jp/arch/jmadata/data/gpv/) during 2007–2016 provided by the Japan Meteorological Agency with the resolution of 1/16°×1/20°, adopting the drag coefficient of Large and Pond (1981). The daily shortwave radiation was based on the newly released of Japanese Ocean Flux Data Sets with Use of Remote Sensing Observation (J-OFURO3) (https://j-ofuro.scc.u-tokai.ac.jp/) with a resolution of 1/4°×1/4° and averaged during 2002 to 2013. The daily longwave, sensible heat flux and latent heat flux were calculated and averaged by adopting bulk formula (Gill, 1982) using hourly air temperature, sea surface temperature, relative humidity, cloud cover, and wind velocity from the GPV-MSM (2007–2016). Daily evaporation was obtained by calculating the latent heat flux. The daily precipitation was provided by the GPV-MSM and averaged hourly from 2007 to 2016. In the study, we want to explore the sensitivity and response of INCWM to sea surface forcing changes, therefore, multi-year average sea surface atmospheric forcing was first used to simulate (named as Control), and then conducted several numerical experiments to analyze the response of INCWM to sea surface forcing changes. Model validations of residual current, water temperature, and salinity in the Seto Inland Sea of Control have been reported in previous studies (Chang et al., 2009; Yu et al., 2016; Zhu et al., 2019). The simulated seasonal variability of INCWM was shown in Section 3.1.

To investigate the response of the INCWM to sea surface forcing changes, we conducted four numerical sensitivity experiments (Table 1) in Section 4.1. The Seto Inland Sea loses heat in winter and gains heat in summer. To determine the

effects of local air-sea heat flux at different stages of the INCWM (based on results of observation data), we increased the air-sea heat flux loss/gain over Iyo-Nada (light blue dotted box in Fig. 1b) by 10 % from previous December to February (cooling season) in Case 1 and from May to July (warming season) in Case 2. In addition, to determine the effects of remote air-sea heat flux change, we increased the air-sea heat flux over the Hayasui Strait (light blue dotted box in Fig. 1b) in July by 10 %in Case 3 and that from May to July by 10 % in Case 4.

**Table 1: List of numerical experiments and conditions.**

| Experiments | Conditions |
| --- | --- |
| Control | climatological sea surface forcing |
| Case1 | The surface heat loss in Iyo-Nada in the previous winter increased by 10 % |
| Case2 | The surface heat gain in Iyo-Nada from May to July increased by 10 % |
| Case3 | The surface heat gain in Hayasui Strait in July increased by 10 % |
| Case 4 | The surface heat gain in Hayasui Strait from May to July increased by 10 % |

## 3 Results

### 3.1 Seasonal variability of the INCWM

The observed spatial distributions of multi-year (1994-2015) averaged water temperature along a transection (Station R to Station 7) from January to December are shown in Fig. 2a-l. The water is well mixed in winter, while thermocline occurs during spring and summer and vanishes in fall in Iyo-Nada (Sta. 1-7). However, water temperature at Hayasui Strait is almost vertically homogeneous throughout a year (Fig. 2a-l, Fig. S1). Seasonal bottom cold water exists locally in the Iyo-Nada. Overall, the INCWM exhibits a prominent seasonal evolution that forms in April, matures in July, and disappears in October. The seasonal cycle is consistent with that of other BCWMs (Horsburgh et al., 2000; Zhang et al., 2012; Chen et al., 2018; Zhu et al., 2018). The main part of the INCWM in July is located at stations 1-4 and below a depth of 20 m with a similar shape of bottom-up "bowl" (Fig. 2g). According to the time series of the multi-year (1994-2015) averaged water temperature at the sea surface and 50 m deep for stations 1-4 (Fig. 2m), the water column mixed well from October to March. Meanwhile, the water temperatures at the sea surface and 50 m deep diverged from April to September, during which the water temperature rose at a rate of 3.0 °C month$^{-1}$ at the sea surface and 1.8 °C month$^{-1}$ at 50 m deep. The water temperature difference between the sea surface and 50 m deep reached 5.5 °C in July. Then, enhanced wind and surface cooling break this stratification in the fall, and consequently, the INCWM disappears in October.

For the model results related to the INCWM, spatial distributions of simulated averaged water temperature in the vertical transection (Fig. S2) show a distinct seasonal variation of INCWM which is similar with observation (Fig. 2a-l). Fig. 2m shows the time series of the simulated water temperature at the sea surface and 50 m deep for stations 1-4. It is likely that the model captured the seasonal evolution of the INCWM given by the observed data up to August. In addition, simulated water

temperature differences between surface and bottom layers from April to September are shown in Fig. 3, which could indicate the position of INCWM. The water temperature difference between surface and bottom layers in July was around 5°C - 6 °C at the central area of Iyo-Nada (Fig. 3d) which corresponds to the location of observed INCWM (Fig. 1b and Fig. 2g). For the horizontal distribution of INCWM, the spatial distribution of water temperature at 50 m depth is presented in Fig. 3g.

On the west side of INCWM, there is a bottom temperature front with a temperature difference (around 2°C) between INCWM and the water on the west side. The INCWM gradually disappears in September (Fig. 2m and Fig. 3f) both in observation and simulation. However, the difference between the simulated and observed water temperatures in the vertical transection increased in September and October (Fig. 2m). The reason for it is that the model is driven by the multi-year average daily wind field, which cannot fully represent episodic strong wind events, especially typhoons, which are the most

active in September (http://www.data.jma.go.jp/obd/stats/data/bosai/tornado/stats/monthly.html). Therefore, the kinetic energy input to the ocean in this simulation (Fig. S2) is much lower than the realistic situation (Fig. 2a-l) during this period. However, our simulation target is the formation and maintenance of the BCWM from the previous winter to summer, which is only slightly related to the model results in September and October. In general, this model captures the main characteristics and seasonal evolution process of INCWM.

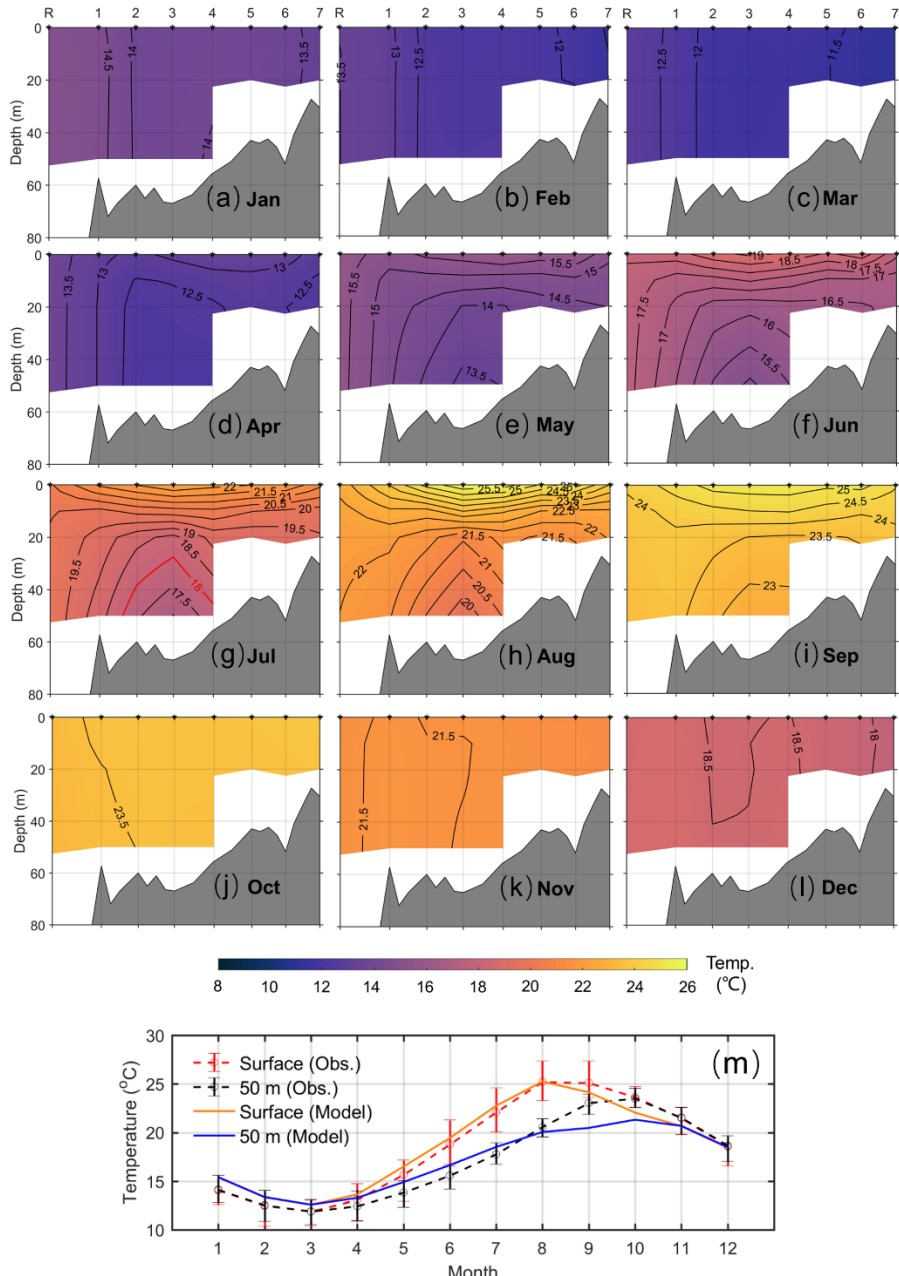

**Figure 2: Spatial distributions of multi-year (1994-2015) averaged water temperature along a vertical transect (station R to station 7) from January to December (a-l). (m) Time series of observed and simulated water temperatures averaged for stations 1-4 at the surface and 50 m depth. The red line of the 18 °C isotherm in (g) is used to define the Iyo-Nada bottom cold water mass (INCWM). The red and black bars in (m) represent the ranges of observed water temperatures at the surface and 50 m depth, respectively.**

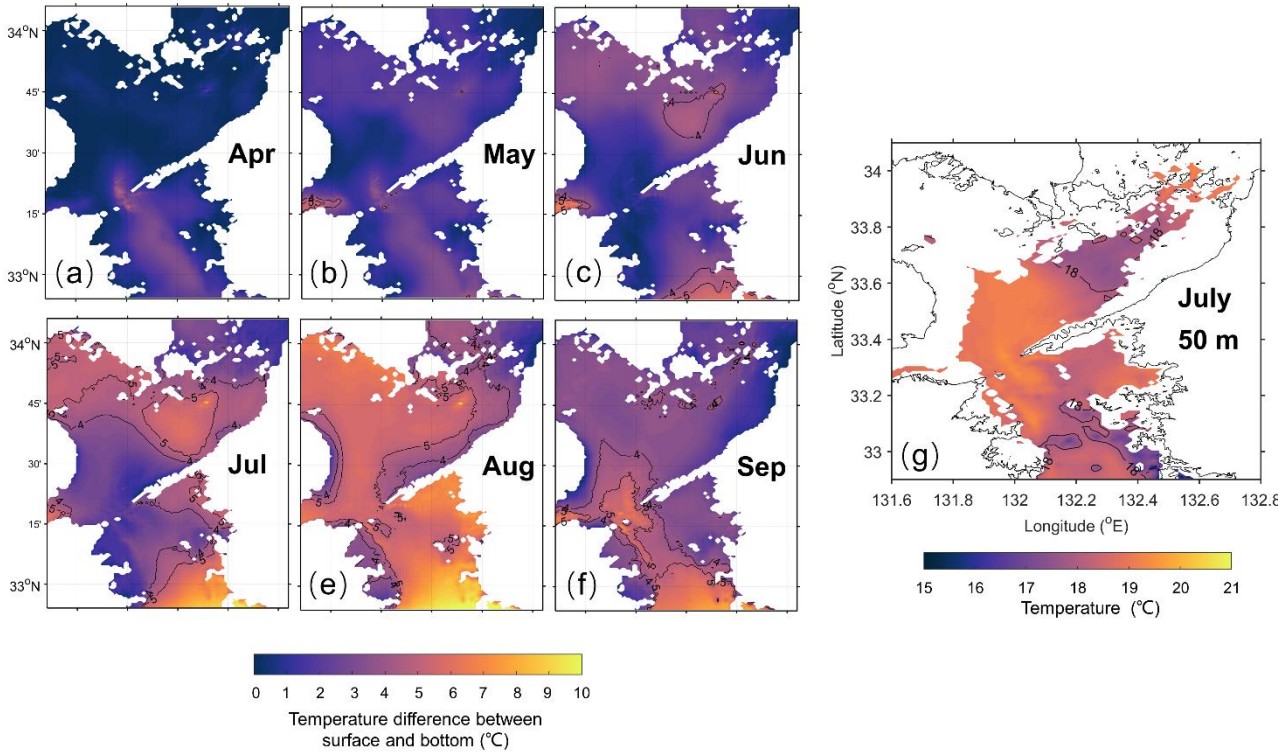

**Figure 3: Simulated temperature differences between surface and bottom from April to September (a-f) and the spatial distribution of water temperature at 50 m depth (g).**

### 3.2 Interannual variations of the INCWM in July

The temperature is lower inside the INCWM than in its surrounding waters in summer. Spatial distributions of water temperature in July along the vertical transection in Iyo-Nada (Fig. 1b) were shown in Fig. S3. The water body below 20 m in the central Iyo-Nada was occupied by cold water around 17-19 ℃ while sea surface temperature was high with the variation of 20 ℃ - 25 ℃. The lowest observed temperature ranged from 15.74 ℃ (2006) and 18.23 ℃ (2007) during 1994-2015(Table S2). A fixed isotherm is often used as the boundary of a cold water mass (Zhang et al., 2012; Yang et al., 2014;
Chen et al., 2018). Observation (Fig. S3) shows that INCWM occupies the central area of Iyo-Nada below 20 m depth with a similar shape of bottom-up "bowl" which can be indicated by the 18 ℃ isotherm in almost all years. In addition, INCWM in July can be well distinguished from surrounding waters by the water temperature difference between surface and bottom layers of 5 ℃ (Fig. 3d) whose location is consistent with that of 18 ℃ isotherm at 50 m depth (Fig. 3g). Therefore, we used the 18 ℃ isotherm to define the boundary of INCWM in July (Fig. 2g), and calculated the average water temperature and
area inside the 18 ℃ isotherm. When calculating the average temperature and area of INCWM along the vertical across-section, we first interpolated the observed water temperature into a rectangular mesh grids (0.01° × 1 m) to get a temperature field, and then calculated the area of grids where temperature less than 18 ℃ and average temperature within

these grids. If there was no water temperature less than 18 °C in the vertical transection, the observed temperature at the average location of the INCWM (50 m depth at station 3) as the average temperature and the area is set to zero. Results were shown in Fig. 4 and Table S2, there was a prominent interannual variation in the water temperature and area of the INCWM. The multi-year average temperature of the INCWM was 17.58 °C with a standard deviation of 0.27 °C. Meanwhile, the lowest average water temperature in the INCWM was 17.04 °C in 2006. Overall, consistent interannual variation was observed for the average and lowest water temperature of the INCWM (figure not shown), exhibiting a correlation coefficient of 0.98 (p < 0.01). The mean area of INCWM was $5.73 \times 10^5$ m$^2$ with a standard deviation of $4.35 \times 10^5$ m$^2$. The smallest area was 0 m$^2$ in 2007 and the largest area was $1.46 \times 10^6$ m$^2$ in 2006. A significant negative correlation (r = -0.86, p < 0.01) was found between the average water temperature and the area of the INCWM, indicating that when the water temperature of the INCWM is lower in a year, the area of the INCWM is larger and the INCWM is stronger.

With increased solar radiation in August, sea surface temperature could reach to 23.85°C (2007)-29.48°C (2008). The INCWM also exists in the vertical transect (Fig. S4) with minimum water temperature ranging 18.52°C (1996) – 20.75°C (2014) which is 2°C-3 °C higher than that in July. In order to show the interannual variation of INCWM in August, we calculated the average temperature and area surround by an isotherm of 20°C as Fig. S4 shown. The results also showed significant negative correlation between average temperature and area which is consistent with the relationship in July. However, there was no significant correlation for interannual variation of INCWM characteristics between July and August. As more typhoons pass through Japan in August than in July (http://www.data.jma.go.jp/obd/stats/data/bosai/tornado/stats/monthly.html), we use the INCWM in July to explore the interannual variation of INCWM in this study to avoid the episodic impact of typhoon as far as possible.

Based on observation data in July, the intensity of the INCWM was defined by two indices, i.e., the spatially averaged water temperature inside the INCWM and the area of INCWM along the observational transect (Fig. 1b), according to the method used in Zhu et al. (2018) studying the intensity of Yellow Sea bottom cold water. Here, we also used this method to get the intensity of the INCWM. Briefly, $X$ is set to be any of the two indices, and $X_0$ is the mean value of X from 1994 to 2015. Thus, $\Delta X_i$ is the anomaly of $X_i$ at year $i$:

$$\Delta X_i = X_i - X_0.$$

The mean range of change over the entire study period is denoted by the mean value of the absolute value of $\Delta X_i$:

$$p = \frac{1}{n}\sum_{i=1}^{n} |\Delta X_i|,$$

where $n$ is the total number of data. A strong INCWM year is characterized by a low water temperature and a large area, while a weak INCWM year is characterized by a high water temperature and a small area. When the water temperature anomaly in a year is smaller than its $-0.5p$ and the area anomaly in a year is larger than its $0.5p$, the year is defined as a strong INCWM year. Conversely, when the water temperature anomaly in a year is larger than its $0.5p$ and the area anomaly is smaller than its $-0.5p$, the year is defined as a weak INCWM year. Note that any other situation is defined as a normal

INCWM year. As a result, the INCWM was strong in 1994, 1996, 2006, 2010, 2012, 2013, and 2015, normal in 1995, 1998-2002 and 2005, while weak in 1997, 2003, 2004, 2007, 2009, 2011, and 2014 (Fig. 4 and Table S2).

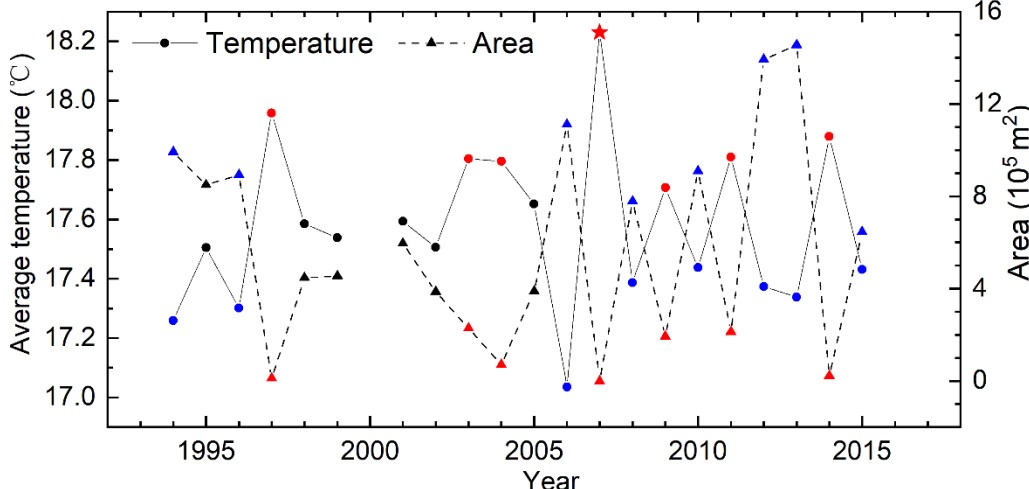

**Figure 4: Time series of mean water temperature (solid line) and vertical cross-section area (broken line) of the Iyo-Nada bottom cold water mass (INCWM) in July during 1994-2015. A strong INCWM year is marked by blue, while a weak INCWM year is**
**marked by red. The red star indicates that we did not identify a BCWM in 2007 because the water temperature was higher than 18 ℃ along the entire transect. For this case, we used the observed temperature (18.23 ℃) at the average location of the INCWM (50 m depth at station 3) as the average temperature and specified a zero area.**

### 3.3 Correlation between INCWM with surrounding waters

As shown in Fig. 2m, the water temperature at the INCWM site was homogeneous in January before gradually forming a distinct dome below a layer warmed by surface heating during spring and summer. To explore the influence of the water temperature at different stages during its formation process on the intensity of INCWM in July, we calculated the correlation coefficients between the average temperature of the INCWM in July (Fig. 4) and the water temperatures at all the depths of each station from 1994 to 2015 for each month (from previous December to July) shown in Fig. 5.


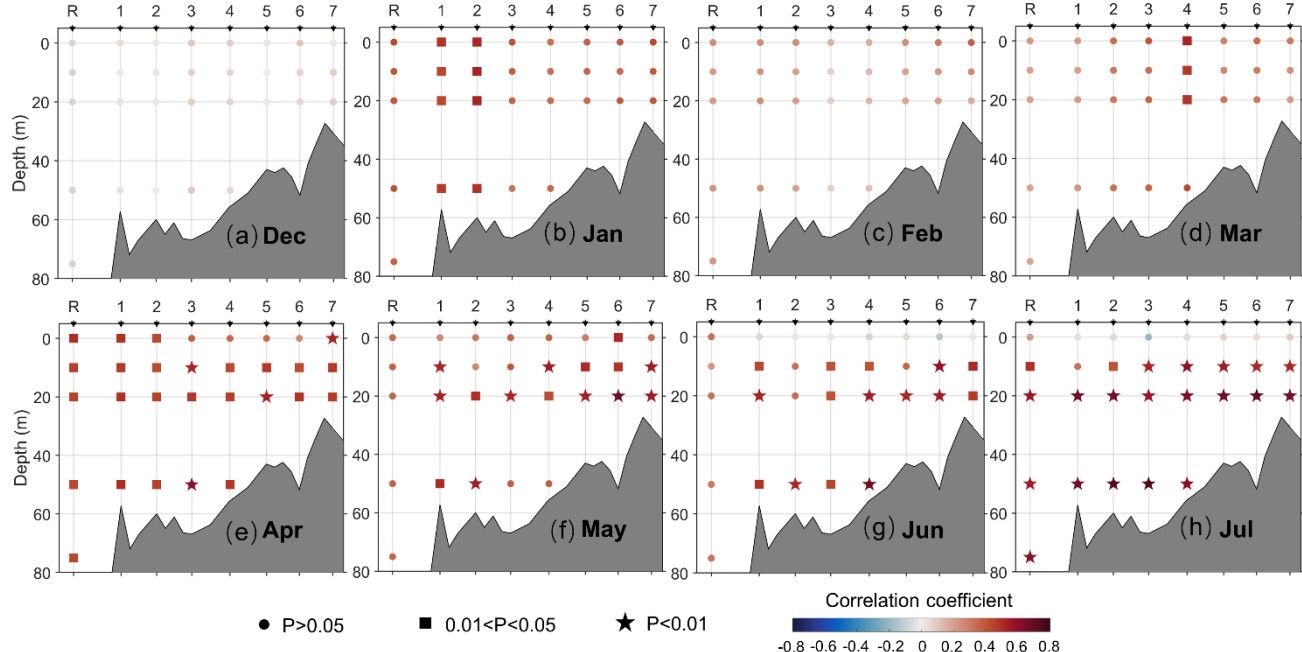

**Figure 5: Correlation coefficients for interannual variation of water temperature at each station from previous December to July (a-h) and that of the Iyo-Nada bottom cold water mass (INCWM) mean water temperature in July during 1994-2015. Circles indicate that the significant confidence level is less than 0.95, squares indicate a significant confidence level is between 0.95 and 0.99, and stars indicate a significant confidence level of more than 0.99.**

Overall, more significant correlations appeared from April to July than from previous December to March (Fig. 5). In April, the water temperatures at almost all stations and depths were significantly correlated with the average water temperature of the INCWM in July (Fig. 5e). From May to July (Figs. 5f-h), although the significant correlations at the sea surface and station R disappeared, the water temperature below 10 m at stations 1-7 still showed significant correlations with the average water temperature of the INCWM in July. The relationship among all the stations and depths was likely enhanced in July, as shown by the correlation coefficients larger than 0.60 (p < 0.01) (Fig. 5h). In addition, a significant correlation below 10 m at station R was observed in July but disappeared in May and June, indicating a distinct connection in July between the interannual variation of the INCWM and the water around station R, which is the Hayasui Strait.

**3.4 Processes influencing the interannual variations of the INCWM**

According to the seasonal evolution of the INCWM, the difference in water temperature between the sea surface and 50 m depth begins in April (Fig. 2m). The water temperature in April was selected as the initial value for the formation of the INCWM in the following months (Fig. 6a). The significant correlations shown in Fig. 5 demonstrate that the local water temperature below 10 m in April is closely related to the interannual variation of the INCWM in July. The water temperature

in April is the result of cooling process in the previous winter. Tsutsumi and Guo (2016) suggested the heat content inside the Seto Inland Sea (except for Bungo Channel and Kii Channel) from January to July mainly depended on air-sea heat fluxes. the initial temperature of the INCWM is likely associated with local air-sea heat flux from winter to early spring.

During the warming season, the effect of local water temperatures below 10 m persists. As Fig. 5e-h shown, the correlation coefficient below 10 m in Iyo-Nada (Sta. 1-7) increased from April to May-July, especially in May and July. In addition, the

significant confidence levels gradually enhanced from $0.01 < p < 0.05$ to $p < 0.01$ during this period, which indicated that heat transport into the INCWM from May to July is also important for the interannual variation of the INCWM in July. Using the temperature difference between the sea surface and 20 m depth at stations 1-4 as the thermocline strength, a significant negative correlation ($r = -0.44$, $p < 0.05$) was obtained for the thermocline strength in July and the average water temperature of the INCWM in July, indicating that a stronger thermocline strength corresponds to a colder INCWM. Because there is no

significant correlation between the INCWM and sea surface temperature at stations 1-5 from May to July (Fig. 5), the stronger thermocline strength acts to reduce the downward heat transport from the sea surface to the INCWM. In addition, river discharge around the Seto Inland Sea increases during the warming season, facilitating the occurrence of an estuarine-like density circulation with the bottom flow from the Hayasui Strait to Iyo-Nada (Fig. 6b), which ultimately promotes horizontal heat transport into the INCWM (Yu and Guo, 2018). When river discharge reaches a peak in July (Zhu et al.,

2019), the bottom horizontal density flow is enhanced, causing more heat to be laterally transported to the INCWM in July than in other months. This process is supported by the significant correlation coefficient of approximately 0.6 ($p < 0.01$) (Fig. 5h) below 10 m at station R in July. Therefore, the lateral heat transport from the Hayasui Strait to Iyo-Nada is another important process that influences the interannual variation of the INCWM in July.

As a summary, the temperature change of INCWM in July on an interannual scale is controlled by three processes, i.e., the

local retention of bottom low water temperature from early spring, local vertical heat diffusion from May to July and horizontal heat advection originating from Hayasui Strait in July (Fig. 6). Since the water temperatures in April and that at Hayasui Strait in July are easy to obtain from observation. The relationship between them with the INCWM intensity is shown in Fig. 7. A strong INCWM corresponds to a combination of a low local initial water temperature in April (11.0-12.5 °C) and a low water temperature at 10-50 m depth at station R in July (19.5-20.3 °C). Conversely, a weak INCWM

usually corresponds to a combination of a high local initial water temperature in April (11.7-14.1 °C) and a high water temperature (19.8-20.9 °C) in the Hayasui Strait at depths of 10-50 m in July. This provides an intuitive understanding about the INCWM intensity and its relationship with initial water temperature before formation and remote horizontal heat advection in July. However, that local vertical heat transfer during warming seasons is not included in Figure 7. As it is not easy to address the relative importance of each process to the interannual variation of the INCWM from only observations,

we would discuss their influence on INCWM characteristics using numerical model in Section 4.1.

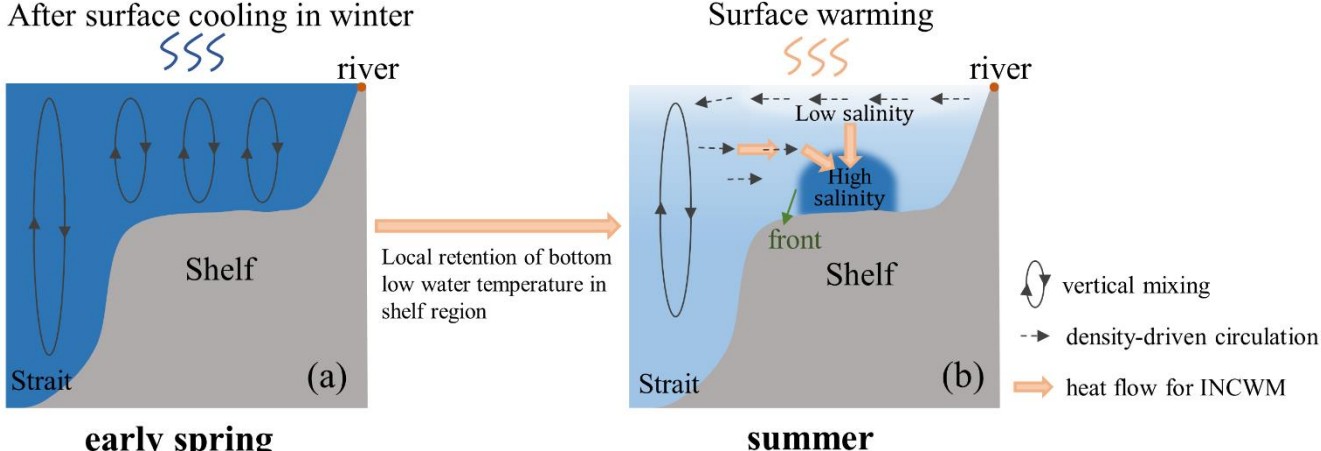

**Figure 6: A schematic diagram of seasonal evolution of INCWM from early spring to summer, which is drawn from references Takeoka (2002), Yu et al. (2016) and Yu and Guo (2018). Low water temperature is indicated by dark blue while high water temperature by light blue.**


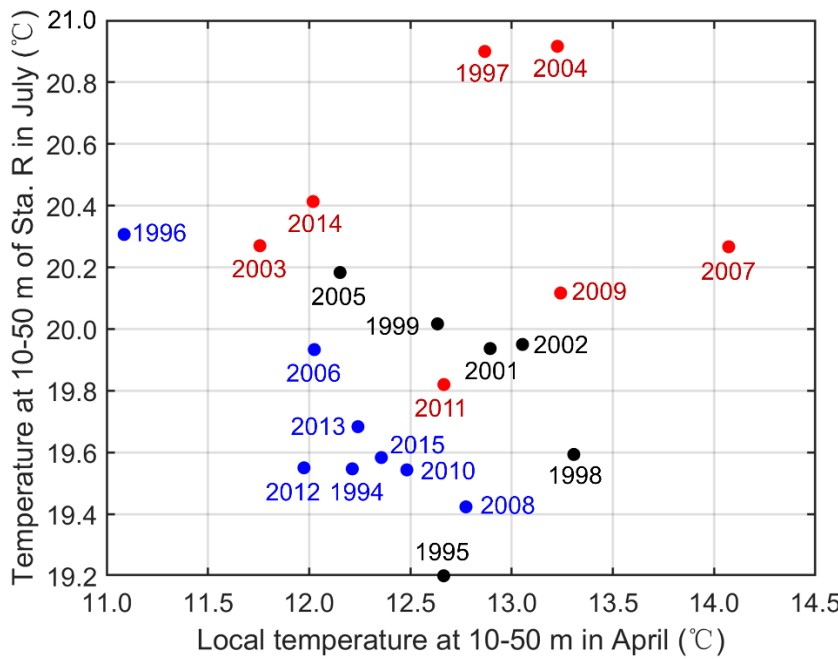

**Figure 7: Scatter plot for the Iyo-Nada bottom cold water mass (INCWM) intensity (color) according to local water temperature below 10 m in April (x-axis) and water temperature below 10 m at station R in July (y-axis). Blue, black, and red solid circles represent strong, normal, and weak INCWM years, respectively, which is consistent with the results found in Fig. 4.**

## 4 Discussion

### 4.1 Sensitivity of INCWM characteristics to air-sea heat flux changes

The interannual variation of water temperature in coastal seas is closely related to sea surface forcing change. Air-sea heat flux is the main heat source to the Seto Inland Sea (Tsutsumi and Guo, 2016) and affects each stage of the INCWM by directly local input or remote input through horizontal heat transport from adjacent areas (Fig. 6). Based on the climatological simulation of the INCWM described in Section 2.2 (Control), we conducted four sensitivity numerical experiments (Table 1) to address the response of INCWM characteristics on local and remote air-sea heat flux changes. According to analyse of observation data (Fig. 6), four factors were designed in the sensitivity experiments which were the local air-sea heat flux in the previous winter (Case 1) and that from May to July (Case 2), remote (Hayasui Strait) air-sea heat flux in July (Case 3) and that from May to July (Case 4).

By these sensitivity numerical experiments, the sensitivity coefficient $q$, defined as the relative change of INCWM characteristic, is defined to quantify the response of the INCWM in July. Using water temperature as an example, $q$ can be calculated using the following formula:

$$q = \frac{T_{case} - T_{control}}{T_{control}}, \tag{1}$$

where, $T_{control}$ is the average water temperature in the area enclosed by a specific isotherm (18 °C, Fig. 2g, July) along the vertical transect (Fig. 2a-l) in the Control run, and $T_{case}$ is the corresponding variable in one of the sensitivity numerical experiments. If there is no water below 18 °C, the water temperature at the mean location of the INCWM is used for $T_{case}$. For the size of the INCWM, $T_{case}$ and $T_{control}$ in Eq. (1) are replaced by the area enclosed by the 18 °C isotherm along the vertical transect (Fig. 2a-l) in the corresponding experiments. If there is no water below 18 °C, the area of the INCWM is set to 0. Except the vertical transection, the $q$ of horizontal INCWM characteristics at 50 m depth layer (mean temperature and area surrounded by the 18°C isotherm) was also calculated to indicate the response of horizontal INCWM. In addition, we investigated the three-dimensional changes of INCWM by calculating its heat content. The sensitivity coefficient of heat content of INCWM ($HC$) was calculated to address the response of heat content on air-sea heat flux changes. $HC$ equals to $\int \rho C_p T dV$, where $C_p$ is the specific heat capacity of seawater (4096 J·kg$^{-1}$·°C$^{-1}$), $\rho$ is the seawater density (1025 kg·m$^{-3}$), $T$ is average water temperature (°C) of INCWM, $V$ is the volume (m$^3$) of INCWM indicated by the 18°C isotherm.

The results of sensitivity numerical experiments showed that the INCWM characteristics changed in all four experiments (Fig. 8 and Table 2). When sea surface water in Iyo-Nada lost more heat in the previous winter in Case 1, the heat content of INCWM in July increased ($q = 0.40$) with the volume of INCWM enlarging ($q = 0.35$ in the vertical plane and $q = 0.13$ in the horizontal plane) and the average temperature decreasing ($q = -1.99 \times 10^{-3}$ in the vertical plane and $q = -1.59 \times 10^{-3}$ in the horizontal plane). In Case 2, When increasing local air-sea heat flux from May to July, the heat content of INCWM in July decreased ($q = -0.92$) along with decreasing volume of INCWM ($q = -1.0$ in the vertical plane and $q = -0.4$ in the horizontal plane). In Case 3, increased remote (Hayasui Strait) air-sea heat flux in July also caused heat content of INCWM decreasing

($q$ = -0.68) along with reduced volume of INCWM. In Case 4, the sensitivity coefficients had little difference with those in Case 3 which suggested that the effect of remote air-sea heat flux change in July is much larger than those in May and June. Comparison between Case 3 and Case 4 confirmed the importance of lateral heat transfer from Hayasui Stratit to Iyo-Nada in July which was also displayed from observation (Fig. 5h). On the whole, the responses of INCWM characteristics in Case 2 and Case 3 were much higher those in Case 1, indicating the importance of air-sea heat flux during stratified season on interannual variation of INCWM.

When changing local air-sea heat flux during the previous winter and warming seasons, the water temperature of INCWM at 50 m depth in July changed around -0.2℃ and 0.4℃ in Case 1 and Case 2, respectively (Fig. 8a, b). However, there seemed a limit of heat transfer around bottom front area (Fig. 8a, b). However, when changing the remote air-sea heat flux in Hayasui Strait, apparent heat transfer occurred across the bottom front area from the west side into INCWM (Fig. 8c). As we changed the same proportion of air-sea heat flux in the four sensitivity experiments, these results suggested that horizontal heat transfer (Fig. 6b) caused by the changes of remote air-sea heat flux in July was an important dynamic process when addressing the dynamic mechanism of interannual variation of INCWM. Nevertheless, Comparing the responses of INCWM in Case 2 and Case 3, they were similar but larger $q$ occurred in Case 2 than those in Case 3 (Fig.8 and Table 2). This meant that INCWM was more sensitive to local air-sea heat flux change from May to July than remote air-sea heat flux change in July.

In terms of INCWM changes in the vertical and horizontal plane, the $q$ in the vertical plane was larger than that in the horizontal plane (Table 2). This suggested that INCWM characteristics in the vertical plane were more likely to change when surface net heat flux changed. Compared the vertical temperature distribution from May to July between Case 2 and Control (Fig. S5), when local sea surface heating increased by 10% in Case 2, the temperature difference at surface ranged from 0.2 ℃ to 0.5 ℃, while the temperature difference inside the INCWM was 0.05 ℃ - 0.1 ℃. Meanwhile, the 0.1 ℃ isotherm of temperature difference (Case 2 - Control) always appeared at around the depth of 10-20 m where the thermocline develops from May to July (Fig. 2e-g). Apparently, more heat stayed above thermocline while the INCWM mainly obtained heat from waters around 10-20 m. From the four sensitivity numerical experiments, we concluded that INCWM characteristics were more sensitive to air-sea heat flux changes during warming seasons than those in the previous winter. Besides, the vertical heat transport process influenced by local air-sea heat flux change in summer seemed more important than horizontal heat advection process influenced by the same change of remote air-sea heat flux in July.

**Table 2: Sensitivity coefficient of INCWM characteristics in four sensitivity numerical experiments.**

| Experiments | Sensitivity coefficient $q$ | | | | |
|---|---|---|---|---|---|
| | Vertical transection | | Horizontal 50 m depth | | 3-D space |
| | Temp. | Area | Temperature | Area | Heat Content |
| Case 1 | $-1.99\times10^{-3}$ | 0.35 | $-1.59\times10^{-3}$ | 0.13 | 0.40 |

| | | | | |
|---|---|---|---|---|
| Case 2 | 2.47×10⁻² | -1.00 | -8.39×10⁻³ | -0.40 | -0.92 |
| Case 3 | 1.40×10⁻² | -0.80 | -1.41×10⁻³ | -0.23 | -0.68 |
| Case 4 | 1.51×10⁻² | -0.82 | -1.45×10⁻³ | -0.23 | -0.68 |

Note: As the vertical area of INCWM is 0 in Case 2, the sensitivity coefficient of the area in the vertical transection is 1.00.

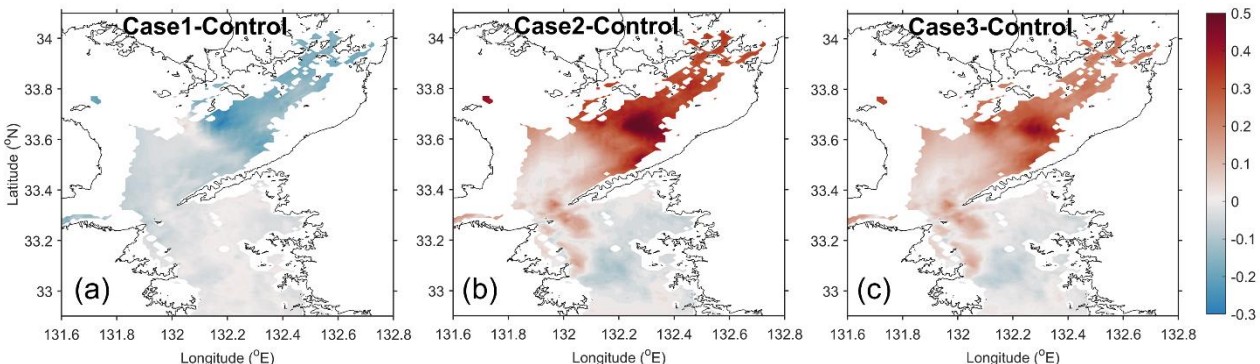

**Figure 8: The spatial distribution of temperature difference (℃) between sensitivity numerical experiments and Control run at 50 m depth in July.**

## 4.2 Comparison with BCWMs in other coastal seas

We investigated the interannual variation of a BCWM in a semi-enclosed coastal sea in Japan using long-term observations and a hydrodynamic model. As previously mentioned, BCWMs have been reported in continental seas worldwide. Overall, most of their seasonal evolutions and formation mechanisms, for example, being a part of remnant water from the previous winter, are similar (Horsburgh et al., 2000; Zhang et al., 2012; Lentz, 2017; Yu and Guo, 2018). Thus, we compared the similarities and differences among several coastal sea BCWMs.

The interannual variations of the Middle Atlantic Bight Cold Pool and Yellow Sea Cold Water Mass and their influencing factors have been described by Chen and Curchitser (2020) and Zhu et al. (2018). In the same way as in these BCWMs, the coherent negative relationship between water temperature and size of BCWM was preserved in our study area, indicating a strong BCWM year featuring a low water temperature and a large size (area or volume). Therefore, as an essential feature of a BCWM, its water temperature and size (area and volume) experience a consistent relationship among different coastal seas, presenting an inherent property that the water temperature inside the BCWM has a negative correlation with the spatial scale of the BCWM over an interannual time scale. Specifically, when heat enters the BCWM from the surrounding water, the water temperature inside the BCWM increases, causing the boundary of the BCWM defined by a specific water temperature to shrink.

Although similarly distinct interannual variations have been observed for these three BCWMs (Yellow Sea Cold Water Mass, Middle Atlantic Bight Cold Pool, and INCWM), their responses to air-sea heat flux change are different. Zhu et al. (2018) suggested that interannual variation in the Yellow Sea Cold Water Mass in summer depends mainly on the water temperature

in the previous winter, which is controlled by local air-sea heat flux during the cooling season. Chen and Curchitser (2020) suggested that the temperature anomaly during winter and spring (mid-January to March) is the primary factor responsible for the interannual variability of the Middle Atlantic Bight Cold Pool temperature during the stratified season. Therefore, for these BCWMs, the initial temperature associated with the air-sea heat flux during the cooling season is important to their interannual variation in summer. However, in this study, we found that INCWM was more sensitive to air-sea heat flux

change in summer than that in the previous winter. It is suggested that influence factors for interannual variations of BCWMs vary in different coastal seas though their seasonal cycles and formation processes are similar. This shows the unique response of different shelf seas to sea surface forcing change. Since BCWMs have important effects on ecosystem and fishery, this finding also provides an insight to understand the different interannual changes of ecosystem and fishery in coastal seas with BCWMs.

Furthermore, we tried to explore the relationship among different BCWMs to help understand the response of bottom water in coastal seas to sea surface forcing change from a comprehensive perspective. Interannual variation in BCWM temperature consists of two parts: the initial temperature after the cooling season, and the increasing range during the warming season. Because the water column vertically mixes in winter, the initial temperature of the BCWM is closely related to the local air-sea heat flux during the cooling season (Chen et al., 2016; Zhu et al., 2018). The increasing range of water temperature

inside the BCWM during the warming season is controlled by the heat input from the surrounding water.

Based on above results, we discuss the water temperature of BCWMs from an idealized model. Because the horizontal scale of BCWM is much larger than vertical scale, as an approximation, we assume that a BCWM has the simple form of a cylinder with radius $R$ (i.e., horizontal size of BCWM) and height H (i.e., average thickness of BCWM). The initial water temperature of the BCWM is homogenous and the vertical heat diffusion coefficient is a constant. Further, during the

warming season, the heat exchange flux with the surrounding water caused by sea surface heat flux is a constant $m$, and that caused by lateral heat flux is a constant $n$. Thus, on a seasonal scale, the following two equations can be established according to the conservation principle of heat:

$$\rho V C_p \frac{dT_1}{dt} = m \cdot \pi R^2 , \tag{2}$$

$$\rho V C_p \frac{dT_2}{dt} = n \cdot 2\pi R \cdot H , \tag{3}$$

where $T_1$ is the BCWM temperature change caused by sea surface heat flux, $T_2$ is the BCWM temperature change caused by lateral heat flux. $V$ is the volume of the cylinder ($V = \pi R^2 H$). Using Eq. (2) and Eq. (3), we can obtain the following relationship:

$$\frac{dT_1}{dt} \propto \frac{m}{H} , \frac{dT_2}{dt} \propto \frac{2n}{R} , \tag{4}$$

Equation (4) reveals that temperature change in the BCWM caused by sea surface heat flux is inversely proportional to $H$ and that caused by lateral heat flux is inversely proportional to $R$. If we use $T$ to represent the temperature of the BCWM, then its variation during the warming season can be given as:

$$\frac{dT}{dt} = \frac{dT_1}{dt} + \frac{dT_2}{dt} = \frac{1}{\rho C_p}\left(\frac{m}{H} + \frac{2n}{R}\right),$$ (5)

$\frac{dT}{dt}$ is inversely proportional to the size of BCWM ($H$ and $R$), suggesting that the larger the size of the BCWM, the smaller the BCWM temperature changes during the warming season.

On this basis, we collected the information of five BCWMs (Table 3). The horizontal size of the BCWMs ($R$) varies widely from 40 km×30 km (Iyo-Nada) to 500 km×300 km (Bering Sea), whereas the average thickness of BCWMs ($H$) changes from around 30 m to 50 m. Because the horizontal size of the BCWMs and their variation are much larger than those in the vertical size, the water temperature variation of BCWM depends mainly on $R$.

As listed in Table 3, the water temperature rising rate during the warming season increases with decreasing horizontal BCWM size, which is consistent with the analysis. The importance of local initial water temperature after a cooling season on the BCWM's temperature in summer is enhanced with an increase in the horizontal size of the BCWM. Note that the increasing rate of temperature in the Middle Atlantic Bight Cold Pool is the same as that of the BCWM in the Irish Sea, although the size of the Middle Atlantic Bight Cold Pool is larger (Table 3). Such difference seems to be the result of an along-isobath mean current of 5 cm s$^{-1}$ in the Middle Atlantic Bight that develops with the cold pool (Lentz, 2017), which leads to increased heat advection into the cold pool. For Iyo-Nada in this study, the size of INCWM is small and it is adjacent to a strait and the relatively strong horizontal density current exists between the strait and BCWM in July, thereby increasing the rate of temperature rise in the INCWM during the warming season (Yu et al., 2016).

**Table 3 Characteristics of bottom cold water mass (BCWMs) in five coastal seas.**

| Coastal seas | Horizontal size of BCWM ($km^2$) | Average thickness of BCWM (m) | Temperature rising rate during warming season ($^\circ$C month$^{-1}$) | Influence factors of interannual variation in the BCWM | References |
|---|---|---|---|---|---|
| Bering Sea | 500 km×300 km | 50 | < 0.15 | -- | Goes et al. (2014); Wang and Zhao (2011) |
| Yellow Sea | 300 km×300 km | 40 | approximately 0.2 | Air-sea heat flux in the previous winter | Zhu et al. (2018) |
| Middle Atlantic Bight | 100 km×300 km | 50 | approximately 1.2 | Initial winter temperature and abnormal warming/cooling due to advection during stratified seasons | Lentz (2017); Chen et al. (2018) |
| Irish Sea | 100 km×100 km | 50 | approximately 1.2 | -- | Holt and Proctor (2003) |
| Iyo-Nada | 40 km×30 km | 30 | approximately 1.8 | more sensitive to air-sea heat flux change during warming seasons than that in the previous winter | Yu and Guo (2018); this study |

Considering the influencing factors on the interannual variation of the three BCWMs (Yellow Sea Cold Water Mass, Middle Atlantic Bight Cold Pool, and INCWM), BCWM size may also be a key factor (Table 3). Here, we considered the influence of air-sea heat flux and lateral heat flux on the interannual variation of the BCWM temperature according to Eq. (5). For a BCWM of a certain size ($H$ and $R$ are constant), the interannual variation of $\frac{dT}{dt}$ during the warming season ($\Delta \frac{dT}{dt}$) is calculated as follows:

$$\Delta \frac{dT}{dt} = \frac{1}{\rho C_p}\left(\frac{\Delta m}{H} + \frac{2\Delta n}{R}\right) = \frac{1}{\rho C_p} \cdot \frac{\Delta m \cdot R + 2\Delta n \cdot H}{HR},\tag{6}$$

where $\Delta m$ is the variation value of air-sea heat flux on an interannual scale, and $\Delta n$ is the variation in the lateral heat flux on an interannual scale. Note that the influence of $\Delta m$ is increased by $R$, while the influence of $\Delta n$ is increased by $H$. As $\Delta n$ is not easily to be evaluated, we could not evaluate the importance of $\Delta m \cdot R$ and $\Delta n \cdot H$, though $R$ is much larger than $H$ (at least 1000 times). Here, we discussed the role of air-sea heat flux during the warming season ($\Delta m$) on the interannual

variations of the five BCWMs. .

According to Eq. (6), the temperature change in a BCWM caused by the interannual variation of air-sea heat flux is proportional to $\frac{\Delta m}{H}$ (W m$^{-3}$), which can be used to compare the contribution of air-sea heat flux during the warming season to the interannual variation of BCWMs in different coastal seas. Monthly air-sea heat flux (total of surface net solar radiation, surface net thermal radiation, surface latent heat flux, and surface sensible heat flux) data during 1979-2020 from the ERA5

dataset (https://www.ecmwf.int/en/forecasts/datasets/reanalysis-datasets/era5, spatial resolution of 0.25°×0.25°) were used in the calculation. We used the standard deviation of air-sea heat flux during the warming season (from May to July) to address $\Delta m$ (Fig. 9). Overall, $\Delta m$ in the five coastal seas ranged from 7.0 W m$^{-2}$ (Yellow Sea) to 11.4 W m$^{-2}$ (Middle Atlantic Bight) (Fig. 9a). After the adjustment of BCWMs thickness ($H$), the value of $\frac{\Delta m}{H}$ (approximately 0.27 W m$^{-3}$) was the biggest in Iyo-Nada, followed by the Middle Atlantic Bight (approximately 0.23 W m$^{-3}$), and the smallest value (approximately 0.17 W m$^{-3}$)

occurred in the Yellow Sea (Fig. 9b). It has been suggested that the air-sea heat flux during the warming season is more sensitive for the interannual variation of the INCWM than that of the Middle Atlantic Bight Cold Pool and the Yellow Sea Cold Water Mass. Thus, a distinct effect of the BCWM thickness was addressed, especially for the INCWM, which has a thin $H$ (approximately 30 m) (Table 3) and small $\Delta m$ (Fig. 9a). Regarding the interannual variation of the Yellow Sea Cold Water Mass, the contribution of air-sea heat flux during the warming season was the smallest among the five BCWMs,

suggesting the importance of local initial water temperature, which is supported by the results of previous studies (Wei et al., 2010; Li et al., 2015; Zhu et al., 2018). For the BCWMs in the Bering Sea and the Irish Sea, it was inferred that the contribution of air-sea heat flux during the warming season is between that of the Yellow Sea and the Middle Atlantic Bight (Fig. 9). In the evaluation, local air-sea heat flux is key factor for interannual variation of INCWM. However, correlation analysis from observation (Fig. 5f-h) shows that the water temperature of INCWM in July is closely related to the water

temperature around the depth of 10-20 m, but is not correlated with sea surface temperature which responses to air-sea heat

flux first. On the one hand, due to the presence of a strong thermocline in summer, the response of INCWM temperature is obviously slower than that at sea surface when local air-sea heat flux changes (Fig. S5). On the other hand, we do not consider the changes in horizontal density-driven current in the above evaluation (Fig. 9b) which is mainly influenced by river discharge and the water temperature in Hayasui Strait. The lateral heat transport could readjust the temperature change

of the INCWM, weakening the consistency between changes of INCWM temperature and sea surface temperature. As for the relative importance of vertical heat diffusion and horizontal heat advection during the warming season, it depends closely on local features, including the topography of each coastal sea, and therefore must be clarified case by case.

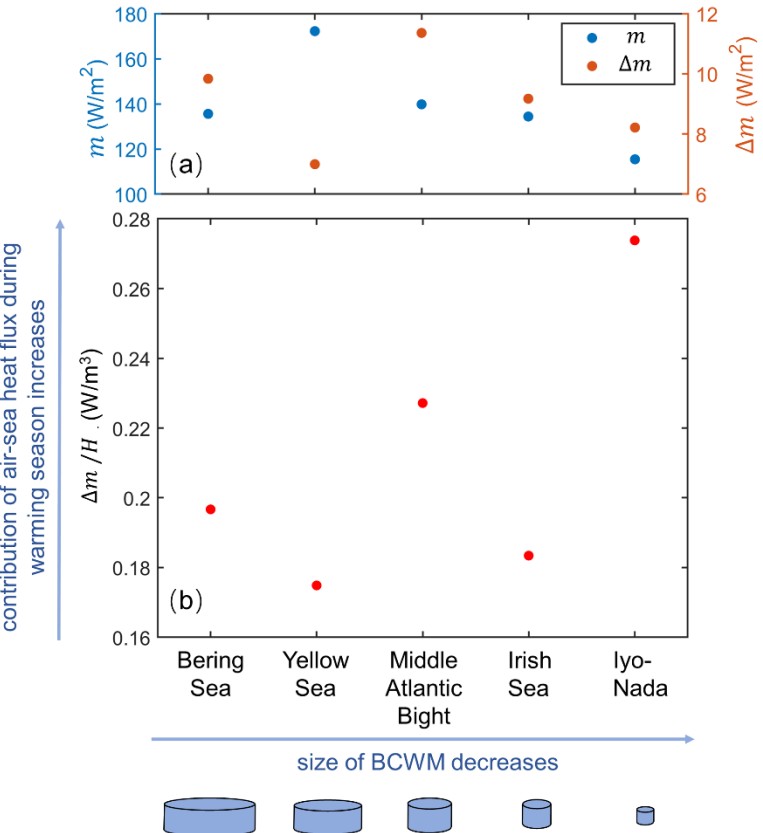

**Figure 9: Values of (a) air-sea heat flux during warming season ($m$) and corresponding interannual variation ($\Delta m$), and (b) $\frac{\Delta m}{H}$ values in the five coastal seas. The cylinder at the bottom means the size of BCWM for which the horizontal and vertical are not proportional.**

**5 Conclusions**

In this study, we investigated the interannual variation of the INCWM from 1994 to 2015 using observational data along a transect across the INCWM and further analysed the response of INCWM characteristics to air-sea heat flux changes. Observation shows that the INCWM is not significant in 2007 with water temperature higher than 18 °C. The INCWM was strong in the years of 1994, 1996, 2006, 2010, 2012, 2013, and 2015, while weak in the years of 1997, 2003, 2004, 2007, 2009, 2011, and 2014. The interannual variation of the mean water temperature inside the INCWM and that of its area show a significant negative correlation. The interannual variation of water temperature inside the INCWM depends on both the water temperature in April (initial temperature) and the vertical and horizontal heat transfer into the INCWM during warming seasons. A strong INCWM corresponds to a low local water temperature in April and a low water temperature in the Hayasui Strait in July. Sensitivity numerical experiments showed that the air-sea heat flux change during the warming season plays an important role in the interannual variation of the INCWM its sensitivity coefficient is larger than that in the previous winter. This means that the heat transport process during the warming season impacted by air-sea heat flux change might be the priority with respect to interannual variation in the INCWM. Conversely, for the Yellow Sea Cold Water Mass, the initial temperature after the cooling season is more important than heat transport during the warming season. This difference is likely related to the size of the BCWMs. A larger BCWM has a weaker dependence on heat transport during the warming season, but a stronger dependence on the initial water temperature before the warming season. This simple relationship will be helpful for understanding the responses of bottom water in different coastal seas to changes in atmospheric forcing.

Our study examined the interannual variation of INCWM characteristics for the first time and evaluated its sensitivity to air-sea heat flux change by a hydrodynamic model. The three processes (Fig. 6) influencing INCWM characteristics changes are discussed qualitatively here. As a future work, a long-term simulation with realistic forcing would be applied to investigate the detail dynamic mechanism controlling the interannual variation of INCWM. Another issue is the biogeochemical aspects around the BCWMs. As BCWM is a nutrient-rich pool in summer, the transport of nutrient across the BCWMs has considerable effects on the phytoplankton growth around the BCWMs (Takeoka, 2002; Williams et al., 2013; Fu et al., 2018). This is an inverse pathway to the heat transport and is expected to be large in some BCWMs.

*Code availability.* The source code of numerical model used in this study is available on request. Please contact Xinyu Guo (guoxinyu@sci.ehime-u.ac.jp)

*Data availability.* The sea surface forcing used for calculation in Section 4 are from DSJRA-55 product (https://jra.kishou.go.jp/) provided by Japan Meteorological Agency and ERA5 product (https://www.ecmwf.int/en/forecasts/datasets/) provided by ECMWF. The observation data is collected from the Prefectural Fishery Research Centers around the Seto Inland Sea which is available on request. Please contact Xinyu Guo (guoxinyu@sci.ehime-u.ac.jp)

*Author contribution.* Junying Zhu: Conceptualization, Investigation, Methodology, Data curation, Formal analysis, Software, Validation, Visualization, Writing – original draft preparation, Writing – review & editing; Jie Shi: Conceptualization, Investigation, Methodology, Data curation, Formal analysis, Writing – original draft preparation, Writing – review & editing; Xinyu Guo: Conceptualization, Methodology, Resources, Software, Validation, Supervision, Writing – review & editing

*Competing interests.* The authors declare that they have no conflict of interest.

*Acknowledgements.* The authors would like to thank the editor and three anonymous reviewers for their constructive comments. This study was supported by the Key Research and Development Program of Hainan Province (ZDYF2020203) and the initial fund from Hainan University for Research and Development (KYQD(ZR)21002). Guo X. was supported by the Environment Research and Technology Development Fund JPMEERF20205005 of the Environmental Restoration and Conservation Agency of Japan. Zhu J. was partly supported by the Ministry of Education, Culture, Sports, Science and 505 Technology, Japan (MEXT) under a Joint Usage/Research Center, Leading Academia in Marine and Environment Pollution Research (LaMer) Project. We would like to thank Editage (www.editage.cn) for English language editing.

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
