# Peer review of "Interannual variation of a bottom cold water mass in the Seto Inland Sea, Japan"

_Ocean Science, 2021_

## Referee Comment (RC1)

Review of os-2021-96

"Air-sea heat flux during warming season determines the interannual variation of bottom cold water mass in a semi-enclosed bay" by Junying Zhu et al.

**Recommendation:**

Major revision

**Summary:**

The authors investigated the interannual variation of a bottom cold water mass (BCWM) formed in the summer in Iyo-Nada, the Seto Inland Sea, based on ship-based hydraulic observations and a threedimensional hydrodynamic model. The results indicate that the heat transport during the stratification season may affect the interannual variation, in addition to water temperature before the season (so-called pre-conditioning). They also considered that control factors of interannual variations differ depending on the size of BCWMs, through comparison of BCWMs in some regions. These results are important in understanding changes in coastal seas and predicting future changes under climate change. I recommend that the paper be published in Ocean Science after some major revisions. My concerns are listed in *Major comments* below.

**Major comments:**

- 1. As a central result of this paper, Figure 5 shows that the interannual variation of the Iyo-Nada bottom cold water mass (INCWM) depends on both the local water temperature in April (the horizontal axis) and the water temperature at the strait in July (the vertical axis). This means that the INCWM are affected by both the early pre-conditioning and the horizontal heat advection during the stratification season (summer). I think this is the most important result of this paper, but is that okay? If so, it appears to be inconsistent with discussion of Sec. 4.1, which emphasizes sea surface heat flux as the main factor based on model results. I suggest to reconsider the title of the paper, too.
  - (a) P.9 L. 213-220

Please emphasize that important results were obtained indicating that the interannual variation depends on both.

(b) P.11 L.245 we conclude that the vertical and horizontal heat transport processes in the warming

season, rather than the initial condition preserved from the previous winter, are responsible for interannual variation in the INCWM in July.

First, the evaluation of the vertical heat flux seems to be inconsistent with the observational results, and so an explanation to address it is needed. Next, Fig. 5 shows that the initial temperature is also important. I do not think the expression "rather than" is grounded.

(c) P.12 L.281 we found that the vertical and horizontal heating processes during the warming season, rather than the initial temperature before warming, were the dominant factor for interannual variation in the INCWM.

Reconsider this part following the above comment (b).

(d) P.14 Table 3

As explained above, I do not think that the "Main factor" of "Iyo-Nada" is only "Air-sea heat flux during stratified seasons".

(e) P.17 Therefore, with respect to interannual variation in the INCWM, the heat transport process during the warming season is more important than the initial temperature after the cooling season.

As noted above, I do not think that "more important" is well-founded.

- 2. Temperature distribution of the INCWM should be shown, not only for the average, but at least for a strong-INCWM year and a weak-INCWM year. In July of Fig.2a alone, the reader does not know what kind of interannual change have occured as a whole. It is also necessary to explain that the July analysis is sufficient. (The same result can be obtained for August, right?)
- 3. A schematic diagram will help the reader's understanding. I want the figure to include a estuary circulation. It would also be better if the figure could be applied to the discussion of "cylinder" in Sec. 4.2.
- 4. Descriptions of the observation and the model specifications are insufficient. Please enrich the explanations.
  - (a) P.3 Sec. 2.1

It is not enough to explain the observation data only in the first paragraph. Please supplement information on observation methods and accuracy. Are there any documents to refer to?

(b) P.5 Sec. 2.2

Although written in Zhu et al. (2019), this paper should also outline the model. The following

explanations are necessary at least.

- i. Model specifications: horizontal resolution, vertical resolution, region, basic classification of model (hydrostatic model? depth-coordinate model?), settings of tides
- ii. Experimental settings: initial value, integration period, lateral boundary conditions, seasurface boundary conditions (moved from Sec. 2.3 to Sec. 2.2), rivers
- iii. The purpose of using multi-year average (climatological normal) data, instead of actual historical data, for sea surface forcings.
- 5. It is necessary to improve the structure of the paper to make it easier to understand. I think it is better that explanation of the analysis method is moved from Sec. 2 to the result sections. Please consider the following modifications.
  - (a) P.5 The first paragraph

Move it to Sec 3.1.

(b) P.6 Sec. 2.3

Move the explanation of the sensitivity coefficient q to Sec. 4.1, since it is used only there. Also, by moving it after defining INCWM, the explanation will be easier to understand. In addition, I could not follow what cr means. The authors need to brush up the explanation. I think that the explanation of the experimental cases should be moved to Sec. 2.2.

**Minor Comments:**

6. P.1 L.14: The interannual variation in water temperature inside the INCWM showed a negative correlation with the area of the INCWM,

It is difficult to understand what kind of interannual variations has been observed. Please give a brief explanation using rough numbers, such as temperature and volume in strong-INCWM years and weak-INCWM years.

 P.5 L.100 The mean range of change over the entire study period is denoted by the mean value of the absolute value of ΔXi

I think that the standard deviation is usually used, when the magnitude of variation over time is investigated. Explain why you use this definition.

8. P.7 L.156 the average water temperature and area inside the 18 C isotherm were calculated.
Please write the formula for calculating the area-averaged value from the observation data, in

order to show the treatment of the area.

9. P.12 L.266 Compared with these BCWMs,

I think the expression "In the same way as in these BCWMs" is better to indicate that the INCWM has the same characteristics as BCWMs in other regions.

10. P.12 L.285

Clarify the purpose of the analysis in the rest of this section.

11. P.15 L.335 As R is much larger than H (at least 1000 times), the influence of  $\Delta m$  is supposed to be more important than that of  $\Delta n$ .

Is it true? The heat flux fluctuation due to horizontal advection,  $\Delta n$ , can be larger by several orders of magnitude than the air-sea heat flux fluctuation,  $\Delta m$ . And, this sentence seems to be inconsistent with the argument that horizontal advection from the strait is more important for interannual variation than the local sea surface heat flux during the stratification season, based on the results of Fig. 4 and Fig. 5.

12. P.17 L.368 The interannual variation of the mean water temperature inside the INCWM and that of its area show a significant negative correlation.

Explain in detail the interannual variation. (See my comment No.6.)

13. P.17 L.383 As an extension, we analyzed the control processes on interannual variation of water temperature in the five BCWMs reported in the literatures using a cylinder column to represent their shape.

This sentence alone is difficult to understand. It is desirable to add a schematic diagram.

14. P.4 Fig. 2

Add a panel number for each month, such as Fig.2 (a) for January, Fig.2 (b) for April etc.

15. P.6 Table 1

Add the CONTROL experiment to the table.

16. P.7 Fig. 3

The "area" means "vertical cross-section area"? It may be misunderstood like a horizontal area.

17. P.8 Fig. 4

The example marks at the bottom right of the figure should be changed from an open circle and star to a closed circle and star.

18. P.8 Fig. 4

This figure does not plot observation points with the significant difference level of 0.95 or less, right? Since it looks as if there had been no observations, those points should be also indicated (maybe black dots?).

---

## Author Comment (AC1)

**Response to Reviewer #1**

**Review of os-2021-96**

"Air-sea heat flux during warming season determines the interannual variation of bottom cold water mass in a semi-enclosed bay" by Junying Zhu et al.

**Recommendation:**

Major revision

**Summary:**

The authors investigated the interannual variation of a bottom cold water mass (BCWM) formed in the summer in Iyo-Nada, the Seto Inland Sea, based on ship-based hydraulic observations and a threedimensional hydrodynamic model. The results indicate that the heat transport during the stratification season may affect the interannual variation, in addition to water temperature before the season (so-called pre-conditioning). They also considered that control factors of interannual variations differ depending on the size of BCWMs, through comparison of BCWMs in some regions. These results are important in understanding changes in coastal seas and predicting future changes under climate change. I recommend that the paper be published in Ocean Science after some major revisions. My concerns are listed in *Major comments* below.

Thanks for your careful reading and comment. Following your comments, we have finished a comprehensive revision on the original manuscript. Below is a point-to-point response.

**Major comments:**

1. As a central result of this paper, Figure 5 shows that the interannual variation of the Iyo-Nada bottom cold water mass (INCWM) depends on both the local water temperature in April (the horizontal axis) and the water temperature at the strait in July (the vertical axis). This means that the INCWM are affected by both the early pre-conditioning and the horizontal heat advection during the stratification season (summer). I think this is the most important result of this paper, but is that okay? If so, it appears to be inconsistent with discussion of Sec. 4.1, which emphasizes sea surface heat

flux as the main factor based on model results. I suggest to reconsider the title of the paper, too.

Thanks for your careful reading. We analyzed the processes influencing the interannual variations of the INCWM in Section 3.3. According to Fig. 5, we found that the influencing processes are not only the early pre-conditioning, but also the vertical and horizontal heat advection during warming season. Since the temperature in April which means pre-conditioning and that at Hayasui Strait in July (which showed the horizontal heat advection) is easier to obtain than the index for vertical heat advection in summer, so we plotted the Figure 6 to show the relationship between the two factors and INCWM intensity. However, the vertical heat transfer is also important which is not easily indicated by observed water temperature in Fig. 6. We modified the description to avoid this misunderstanding in Section 3.3.

"As a summary, the temperature change of INCWM in July on an interannual scale is controlled by three heat transfer processes, i.e., the local retention of bottom low water temperature from early spring, local vertical heat diffusion from May to July and horizontal heat advection originating from Hayasui Strait in July (Fig. 6). Since the water temperatures in April and that at Hayasui Strait in July are easy to obtain from observation. The relationship between them with the INCWM intensity is shown in Fig. 7. A strong INCWM corresponds to a combination of a low local initial water temperature in April (11.0-12.5 °C) and a low water temperature at 10-50 m deep at station R in July (19.5-20.3 °C). Conversely, a weak INCWM usually corresponds to a combination of a high local initial water temperature in April (11.7-14.1 °C) and a high water temperature (19.8-20.9 °C) in the Hayasui Strait at depths of 10-50 m in July. This provides an intuitive understanding about the INCWM intensity and its relationship with initial water temperature before formation and remote horizontal heat advection in July. However, it is noted to say that local vertical heat transfer during warming seasons is not included in Figure 7."

Global climate change affects coastal seas via regional sea surface forcing. Previous studies had suggested that air-sea heat flux is the key factor for water temperature variation inside the Seto Inland Sea (Tsutsumi and Guo, 2016) and an important factor influencing the variation of other BCWMs (Zhu et al., 2018). Combined with observation results, we discussed the influence of air-sea heat flux to INCWM characteristics using a hydrodynamic model in Section 4.1 and several sensitivity numerical experiments. The Section 4.1 is a further discussion based

on Figure 6 and it is consistent with Figure 6.

In this study, we demonstrated the interannual variation of INCWM and its influence process using long-term observation data, and discussed the sensitivity of INCWM to air-sea heat flux changes. According to your suggestion and our deep thinking, we changed the title to "Interannual variation of a bottom cold water mass in the Seto Inland Sea, Japan" in the revised manuscript.

(a) P.9 L. 213-220

Please emphasize that important results were obtained indicating that the interannual variation depends on both.

Thanks and changed as your suggestion. We demonstrated the importance of not only water temperature before INCWM formation, but also the heat transport processes (horizontal and vertical) during warming seasons for interannual variation of INCWM in the last paragraph in Section 3.3.

(b) P.11 L.245 we conclude that the vertical and horizontal heat transport processes in the warming season, rather than the initial condition preserved from the previous winter, are responsible for interannual variation in the INCWM in July.

First, the evaluation of the vertical heat flux seems to be inconsistent with the observational results, and so an explanation to address it is needed. Next, Fig. 5 shows that the initial temperature is also important. I do not think the expression "rather than" is grounded.

Thanks for your comment. The Fig.5 has been changed as Fig. 6 in the revised manuscript. First, for influencing process on interannual variation of INCWM, we demonstrated the importance of early pre-conditioning, the vertical heat transport during warming season (from May to July) and horizontal heat advection in July according to observation data (Figure 5). Figure 6 just provides an intuitive understanding about INCWM intensity and initial water temperature before formation and horizontal heat advection in summer. We did not compare the relative importance among the three processes based on observation data.

In the old manuscript, we tried to quantify the contribution of sea surface forcing on interannual variation of INCWM using an index (cr). However, cr has no clear physical meaning and is not suitable to assess the contribution rate. Therefore, we remove the information about cr in the revised manuscript. We just evaluated the sensitivity of INCWM to air-sea heat flux changs.

By comparing sensitivity coefficient q in the vertical and horizontal plane, as well as heat content of INCWM change (Table 2), the conclusion was changed as "*From the three sensitivity numerical experiments, we concluded that INCWM characteristics were more sensitive to air-sea heat flux changes during warming seasons than those in the previous winter*" in the last paragraph in Section 4.1.

(c) P.12 L.281 we found that the vertical and horizontal heating processes during the warming season, rather than the initial temperature before warming, were the dominant factor for interannual variation in the INCWM.

Reconsider this part following the above comment (b).

Refer to the answer for the previous question, according to sensitivity numerical experiments, we have changed the sentence as "*INCWM was more sensitive to air-sea heat flux change in summer than that in the previous winter*" in the third paragraph in Section 4.2.

(d) P.14 Table 3

As explained above, I do not think that the "Main factor" of "Iyo-Nada" is only "Air-sea heat flux during stratified seasons".

We reconsidered the influence factor based on the results of sensitivity numerical experiments and changed as "*more sensitive to air-sea heat flux change during warming seasons than that in the previous winter*" in Table 3.

(e) P.17 Therefore, with respect to interannual variation in the INCWM, the heat transport process during the warming season is more important than the initial temperature after the cooling season. As noted above, I do not think that "more important" is well-founded.

As answered above, we changed as "Sensitivity numerical experiments showed that the air-sea heat flux change during the warming season plays an important role in the interannual variation of the INCWM its sensitivity coefficient is larger than that in the previous winter. This means that the heat transport process during the warming season impacted by air-sea heat flux change might be the priority with respect to interannual variation in the INCWM" in the first paragraph in Section 5. 2. Temperature distribution of the INCWM should be shown, not only for the average, but at least for a strong-INCWM year and a weak-INCWM year. In July of Fig.2a alone, the reader does not know what kind of interannual change have occured as a whole. It is also necessary to explain that the July analysis is sufficient. (The same result can be obtained for August, right?)

Thanks for your suggestions. We add the temperature distribution of the INCWM in July from 1994 to 2015 in the Supplementary Fig. 3 and characteristics of INCWM in July in Supplementary Table 2. Meanwhile, we add the relevant description about the interannual variation of INCWM in Section 3.1.

In August, the INCWM also exists in the vertical transect (Supplementary Fig. 4) with minimum water temperature ranging 18.52°C (1996) – 20.75°C (2014) which is 2°C-3 °C higher than that in July. We calculated the average temperature and area surround by an isotherm of 20°C as Supplementary Fig. 4 shown. The results also showed significant negative correlation between average temperature and area which is consistent with the relationship in July. However, there was no significant correlation for interannual variation of INCWM characteristics between July and August. Because more typhoons pass through Japan in August than in July (http://www.data.jma.go.jp/obd/stats/data/bosai/tornado/stats/monthly.html), we use the INCWM in July to explore the interannual variation of INCWM in this study to avoid the episodic impact of typhoon as far as possible. The information about INCWM in August has added in the second paragraph in Section 3.1.

 A schematic diagram will help the reader's understanding. I want the figure to include a estuary circulation. It would also be better if the figure could be applied to the discussion of "cylinder" in Sec. 4.2.

Thanks for your suggestion. We plotted the schematic diagram (below) of seasonal evolution of INCWM from the previous winter to this summer according to Takeoka (2002), Yu et al. (2016) and Yu and Guo (2018) as Fig. 6 in the revised manuscript. In summer, a density-induced gravitational circulation occurs as the bottom water flows from the Hayasui Strait to Iyo-Nada, whereas the surface water flows in the opposite direction. This circulation can be enhanced in July by an abrupt increase in river discharge into the Seto Inland Sea. The figure helps to understand the heat transport process about INCWM.

In Section 4.2, we simplified BCWM as a cylinder as the horizontal scale of BCWM is much

larger than vertical scale. However, the vertical scale is enlarged in the schematic diagram for appropriate presentation (Fig. 6). The schematic diagram is not suitable for the discussion of "cylinder" in Section 4.2. We showed the "cylinder" in Fig. 8 to present the discussion result in Section 4.2, which means that the size of INCWM is an important factor for investigating interannual variation of BCWMs.

Figure A schematic diagram of seasonal evolution of INCWM from early spring to summer, which is drawn from references Takeoka (2002), Yu et al. (2016) and Yu and Guo (2018). Low water temperature is indicated by dark blue while high water temperature by light blue.

4. Descriptions of the observation and the model specifications are insufficient. Please enrich the

explanations.

(a) P.3 Sec. 2.1

It is not enough to explain the observation data only in the first paragraph. Please supplement information on observation methods and accuracy. Are there any documents to refer to?

The observation data was collected from Ehime Prefectural Fishery Research Centers (https://www.pref.ehime.jp/h35115/ehime-suiken.html), we have added relevant description in the first paragraph in Section 2.1 and Supplementary Table 1. The water temperature was measured by ALEC CTD carried by survey vessel "Yoshuu" ( $L \downarrow \wp \bar{\gamma}$ ). The measurement accuracy is 0.001 degree. We add the information in Section 2.1.

(b) P.5 Sec. 2.2

Although written in Zhu et al. (2019), this paper should also outline the model. The following explanations are necessary at least.

i. Model specifications: horizontal resolution, vertical resolution, region, basic classification of model

(hydrostatic model? depth-coordinate model?), settings of tides

ii. Experimental settings: initial value, integration period, lateral boundary conditions, sea surface boundary conditions (moved from Sec. 2.3 to Sec. 2.2), rivers

iii. The purpose of using multi-year average (climatological normal) data, instead of actual

historical data, for sea surface forcings.

Thanks for your suggestions. We have added the detailed model configuration formation in the first paragraph in Section 2.2.

"Model domain is 130.98°E-135.5°E, 32.8°N-34.8°N covering the entire Seto Inland Sea. The model has a horizontal resolution of around 1 km ( $1/80^{\circ}$  in the zonal direction and  $1/120^{\circ}$  in the meridional direction), and a vertical resolution of 21 sigma levels. The initial temperature and salinity fields in January were produced by merging the Marine Information Research Center dataset in the SIS region and the model results of Guo et al. (2003). The boundary conditions including de-tided current velocity, temperature, salinity, and surface elevation were obtained from the diagnostic model of Guo et al. (2004). Four major tidal constituents (M2, S2, O1 and K1) were considered and the daily river discharges averaged over 24 years (1993– 2016) from the Ministry of Land, Infrastructure and Transport were used in the model. Multiyear averaged daily surface fluxes of momentum, heat and fresh water was used to drive model (Zhu et al., 2019). The daily wind stress was based on hourly averaged results of wind stress, which was calculated by wind velocity from the Grid Point Value of Meso-Scale Model (GPV-MSM) (http:// database.rish.kyoto-u.ac.jp/arch/jmadata/data/gpv/) during 2007–2016 provided by the Japan Meteorological Agency with the resolution of  $1/16^{\circ} \times 1/20^{\circ}$ , adopting the drag coefficient of Large and Pond (1981). The daily shortwave radiation was based on the newly released of Japanese Ocean Flux Data Sets with Use of Remote Sensing Observation (J-*OFURO3)* (https://j-ofuro.scc.u-tokai.ac.jp/) with a resolution of  $1/4^{\circ} \times 1/4^{\circ}$  and averaged during 2002 to 2013. The daily longwave, sensible heat flux and latent heat flux were calculated and averaged by adopting bulk formula (Gill, 1982) using hourly air temperature, sea surface temperature, relative humidity, cloud cover, and wind velocity from the GPV-MSM (2007–2016). Daily evaporation was obtained by calculating the latent heat flux. The daily precipitation was provided by the GPV-MSM and averaged hourly from 2007 to 2016."

5. It is necessary to improve the structure of the paper to make it easier to understand. I think it is better

that explanation of the analysis method is moved from Sec. 2 to the result sections. Please consider the following modifications.

(a) P.5 The first paragraph

Move it to Sec 3.1.

Thanks and changed as your suggestion.

(b) P.6 Sec. 2.3

Move the explanation of the sensitivity coefficient q to Sec. 4.1, since it is used only there. Also, by moving it after defining INCWM, the explanation will be easier to understand. In addition, I could not follow what cr means. The authors need to brush up the explanation. I think that the explanation of the experimental cases should be moved to Sec. 2.2.

Thanks for your suggestions. We tried to explore the sensitivity of INCWM to sea surface forcing changes in Sec. 4.1, according to peer reviews and deep thinking, the *cr* was removed in the revised manuscript because it could not account for the contribution of one factor on interannual variation of INCWM.

The sensitivity coefficient q was used to quantify the response of INCWM to sea surface forcing changes. As your suggestion, we move the explanation of the sensitivity coefficient q to Section 4.1 and move the explanation of the experimental cases to Sec. 2.2.

**Minor Comments:**

6. P.1 L.14: The interannual variation in water temperature inside the INCWM showed a negative correlation with the area of the INCWM,

It is difficult to understand what kind of interannual variations has been observed. Please give a brief explanation using rough numbers, such as temperature and volume in strong-INCWM years and weak-INCWM years.

Response:

Thanks and changed as suggestion. We changed the sentence as "Surrounded by 18 °C isotherm, The observed multi-year average water temperature inside the INCWM was 17.58 °C with a standard deviation of 0.27 °C, while the mean area of INCWM was  $5.73 \times 105 \text{ m}2$  with a standard deviation of  $4.35 \times 105 \text{ m}2$ . Their interannual variation showed a negative correlation with the area of the INCWM that indicates a low temperature corresponds to a big area". 7. P.5 L.100 *The mean range of change over the entire study period is denoted by the mean value of the absolute value of*  $\Delta X_i$

I think that the standard deviation is usually used, when the magnitude of variation over time is investigated. Explain why you use this definition.

Thanks your suggestion. As your suggestion, we have already used the standard deviation to get the intensity of INCWM. Using the standard deviation, we only get one strong INCWM year (2006) and two weak INCWM years (1997 and 2014). The INCWM intensity in other year was not recognized which is not beneficial to analyze characteristic of strong and weak INCWM years in Fig. 5. To widely identify the intensity of INCWM, we used the method described in the paper which have been applied to Yellow Sea Cold Water Mass (Zhu et al., 2018) to get the intensity of INCWM. By this method, the strong INCWM year (2006) and weak INCWM years (1997 and 2014) identified by the standard deviation were also recognized, meanwhile, it recognized the INCWM intensity in other years which is beneficial to analyze characteristic of strong and weak INCWM years in Fig. 7. Therefore, it was better to get the INCWM intensity using this method than the standard deviation.

**8. P.7 L.156 the average water temperature and area inside the 18 C isotherm were calculated.**

Please write the formula for calculating the area-averaged value from the observation data, in order to show the treatment of the area.

Thanks your suggestion. We add the following description of the calculation method in the first paragraph of Section 3.1.

"When calculating the average temperature and area of INCWM along the vertical acrosssection, we first interpolated the observed water temperature into a rectangular mesh grids  $(0.01^{\circ} \times 1 \text{ m})$  to get a temperature field, and then calculated the area of grids where temperature less than 18 °C and average temperature within these grids. If there was no water temperature less than 18 °C in the vertical transection, the observed temperature at the average location of the INCWM (50 m deep at station 3) as the average temperature and the area is set to zero."

9. P.12 L.266 Compared with these BCWMs,

I think the expression "In the same way as in these BCWMs" is better to indicate that the INCWM has

the same characteristics as BCWMs in other regions.

Thanks and changed as suggestion.

**10. P.12 L.285**

Clarify the purpose of the analysis in the rest of this section.

Thanks for your suggestions, we added the sentences "It is suggested that influence factors for interannual variations of BCWMs vary in different coastal seas though their seasonal cycles and formation processes are similar. This shows the unique response of different shelf seas to climate change. Since BCWMs have important effects on ecosystem and fishery, this finding also provides an insight to understand the different interannual changes of ecosystem and fishery in coastal seas with BCWMs.

11. P.15 L.335 As R is much larger than H (at least 1000 times), the influence of  $\Delta m$  is supposed to be more important than that of  $\Delta n$ .

Is it true? The heat flux fluctuation due to horizontal advection,  $\Delta n$ , can be larger by several orders of magnitude than the air-sea heat flux fluctuation,  $\Delta m$ . And, this sentence seems to be inconsistent with the argument that horizontal advection from the strait is more important for interannual variation than the local sea surface heat flux during the stratification season, based on the results of Fig. 4 and Fig. 5. Kindly, based on Fig. 5, we demonstrated the importance of early pre-conditioning, the vertical heat transport during warming season (from May to July) and horizontal heat advection in July. However, we did not compare the relative importance among the three processes based on observation data. As for Fig. 7, we just showed the relationship between INCWM intensity and local water temperature in April, water temperature of Hayasui Strait in July. We changed the description of Fig. 5 and Fig. 7 to avoid this misunderstanding in the revised manuscript. Thanks for your comment. For the sentence,  $\Delta m$  is the variation value of air-sea heat flux on an interannual scale, and  $\Delta n$  is the variation in the lateral heat flux on an interannual scale. Since  $\Delta n$  is not easily to be evaluated, this statement is too arbitrary though R is much larger than H (at least 1000 times). We changed this sentence as "As  $\Delta n$  is not easily to be evaluated, we could not evaluate the importance of  $\Delta m \cdot R$  and  $\Delta n \cdot H$ , though R is much larger than H (at least 1000 times)" in Section 4.2.

12. P.17 L.368 The interannual variation of the mean water temperature inside the INCWM and that of

its area show a significant negative correlation.

Explain in detail the interannual variation. (See my comment No.6.)

Thanks for your comment. We have added the description about the interannual variation of INCWM in the Section 5.

"Observation shows that the INCWM is not significant in 2007 with water temperature higher than 18 °C. The mean water temperature inside the INCWM was 17.58 °C with the lowest temperature 17.04 °C (2006) and the highest temperature 18.23 °C (2007), while the mean area of INCWM was  $5.73 \times 106 \text{ m}2$  with the smallest area 0 m2 (2007) and the largest area 1.46 × 106 m2 (2006). The interannual variation of the mean water temperature inside the INCWM and that of its area show a significant negative correlation"

13. P.17 L.383 As an extension, we analyzed the control processes on interannual variation of water temperature in the five BCWMs reported in the literatures using a cylinder column to represent their shape.

This sentence alone is difficult to understand. It is desirable to add a schematic diagram.

Thanks for your suggestion. We added a schematic diagram of seasonal evolution of INCWM in Fig. 6 and "cylinder column" in Fig. 9 to make it clear. we simplified BCWM as a cylinder as the horizontal scale of BCWM is much larger than vertical scale. In Fig. 9, the cylinder at the bottom means the size of BCWM for which the horizontal and vertical are not proportional. 14. P.4 Fig. 2

Add a panel number for each month, such as Fig.2 (a) for January, Fig.2 (b) for April etc.

Thank you and changed as suggestion.

15. P.6 Table 1

Add the CONTROL experiment to the table.

Thank and changed as suggestion.

16. P.7 Fig. 3

The "area" means "vertical cross-section area"? It may be misunderstood like a horizontal area. Thanks for your suggestion. The "area" means "vertical cross-section area" and we have changed the wording in Fig. 4.

**17. P.8 Fig. 4**

The example marks at the bottom right of the figure should be changed from an open circle and star to a closed circle and star.

Thank you and changed as suggestion.

**18. P.8 Fig. 4**

This figure does not plot observation points with the significant difference level of 0.95 or less, right? Since it looks as if there had been no observations, those points should be also indicated (maybe black dots?).

Thanks for your suggestion. We repainted the Fig. 5 and marked correlation coefficients and significant difference levels at all observation depth of each station.

Figure 5: Correlation coefficients for interannual variation of water temperature at each station from previous December to July (a-h) and that of the Iyo-Nada bottom cold water mass (INCWM) mean water temperature in July during 1994-2015. Circles indicate that the significant difference level is less than 0.95, squares indicate a significant difference level is between 0.95 and 0.99, and stars indicate a significant difference level of more than 0.99.

---

## Author Comment (AC2)

Referee comment on "Air-sea heat flux during warming season determines the interannual variation of bottom cold water mass in a semi-enclosed bay" by Junying Zhu et al., Ocean Sci. Discuss., https://doi.org/10.5194/os-2021-96-RC2, 2021

This paper focuses on the interannual variability of the cold water mass in Iyo-Nada, Japan, using both observations along a transect and three sensitivity runs of a hydrodynamic model (POM). The main conclusion of this study is that it is the heat transport during the warming season (May-July), rather than initial temperature before warming (Dec-Feb) that dominates the cold water mass's interannual variability. In the end, this paper also compares with other coastal cold water masses using idealized equations to explain why the inter annual variability of the INCWM is not dominated by initial temperature in winter.

This is an interesting topic, however, the major conclusion is not successfully delivered through the three sensitivity runs. As Table 1 suggests, the authors increased local and remote air-sea heat fluxes by 10% during the winter (Dec-Feb) and warming (May-July) seasons, in order to evaluate whether it is initial temperature change in winter or heat transports in warming season that dominates the interannual variability of the cold water mass. Design of these sensitivity runs does not separate these two roles at all, thus not convincing at all. Their reasoning is like this, process A + process B will lead to phenomenon C. The authors changed 1.1A kept B the same, and it leads to 1.3C, but the combination of A and 1.1B leads to larger change in C, so B is the dominant role for C, instead of A, which is funny.

For example, authors finally found Case 2 - increase air-sea heat flux by 10% during May- July that brings the largest q and cr(T) values. In this case, the interannual signal in initial temperature is still kept in the model, it is not convincing to conclude as the authors suggested, only if the initial temperature signal on interannual time scale is totally removed. On the other hand, the increase value (10%) is tricky as well. Authors at least need other runs with different increase values to make sure this 10% is not sensitive and the conclusion is still valid. Not to say the results suggested by cr(A) are not supporting, which also need reasonable explanation.

Another confusing and disappointing point is, the authors do use a numerical model to study the interannual variability of this cold water mass, but none of heat budget terms from the model is analyzed, which is super crucial for determining which physical process that is mainly contributing. The authors write many texts for the last part (Section 4.2), which is actually not that interesting. Too many idealized assumptions, e.g. no interannual variabilities in rho, R, and H? And the final conclusion of this part is obvious without these texts.

Therefore, this paper is suggested to modify their reasoning method (three sensitivity runs) and the paper structure (too many uninteresting texts in discussion and few analysis about heat budgets). Other comments are listed below.

Thanks for your careful reading and comments. Following your comments, we have finished a comprehensive revision on the original manuscript. Below is a point-to-point response.

We conducted a climatological simulation in the study. The two roles (initial temperature change in winter or heat transports in warming season) were not separated in the sensitivity runs because the climatological run did not contain the signals of interannual variability. The target of sensitivity runs about "1.1 process A + process B" and "process A + 1.1 process B" is to and explore the sensitivity of INCWM on air-sea heat flux. As your comment, it is not enough to get the dominant role for C, however, it can test for sensitivity. Though changed by 10% of a factor, we investigate the response of INCWM.

In the old manuscript, we try to quantify the contribution (cr) of one factor to interannual variation of INCWM using sensitivity coefficient q and the range of interannual variation of the factor (delta *f*). According to peer reviews and our deep thinking, *cr* has no significant physical meaning and is not suitable to account for the contribution of one factor on interannual variation of INCWM. Therefore, the content about *cr* is removed in the revised manuscript and focus on the sensitivity of INCWM rather than discussing the dominant factor.

After the comparison of different BCWMs with different influencing process for its interannual variation, we tried to explore the relationship among them which is helpful to understand the response of bottom water in coastal seas to climate change from a comprehensive perspective. The idealized derivation helps us to understand theoretically in Section 4.2 and this part is an extension of the study, so we keep this part. The structure of this paper has been reorganized in the revised manuscript.

**Major Comments:**

1. Section 2.1 does not include the time information of the observational dataset. Is it only collected in

Jan/Apr/July/Oct each year for all eight stations? It seems not true, because Figure 2b compares between observational datasets and model results on a time scale of multi-year monthly climatology. Also, How many obs. in each month are collected? Do you calculated monthly averages first? Thanks for your comment. We added the observation year for each month in Iyo-Nada and Hayasui Strait during 1994-2015 (Supplementary Table 1) and the relevant information about observation in Section 2.1. We collected the observation data from January to December and calculated monthly average first. In Fig. 2, We add the vertical spatial distribution of monthly average temperature in the revised manuscript.

Figure 2a. In the text, it says the R station has a depth of 75m, but the figure only shows the upper 50m. In the caption, it says the 18°C isotherm is considered to define the studied cold water mass, which is better to be illustrated in the main text, e.g. the method section.

Thanks for your comment and suggestion. Observation only has a depth of 75 m at R station, while the deepest observation at other stations are around 50 m and 20 m. To avoid unreliable interpolation, we did not show the water temperature at 75 m of station R. In the Section Introduction, we demonstrated that "The depth of Hayasui Strait is more than 100 m and the water temperature is homogenous throughout the year because of strong tidal mixing (Kobayashi et al., 2006)" to illustrate the vertical water temperature at station R. Also, we added the vertical distribution of multi-year average water temperature at station R in Supplementary Fig. 1 to show the seasonal variation of water temperature at Hayasui Strait in Section 2.1.

A fixed isotherm is often used as the boundary of a cold water mass, such as 2 °C for Bering Sea Cold Water Mass (Zhang et al., 2012), 10 °C for the Middle Atlantic Bight Cold Pool (Yang et al., 2014) and 10 °C for Yellow Sea Cold Water Mass (Chen et al., 2018). As for INCWM in this study, we can see obvious bottom cold water below 18 °C occupying the central Iyo-Nada in the spatial distributions of temperature in July from 1994 to 2015 in Supplementary Fig. 3. Therefore, we use 18°C isotherm to define the INCWM in the study. We add relevant illustration in Section 2.1 and Section 3.1 in the revised manuscript.

3. Section 2.2 Model validation: This paper focuses on the interannual variability of the cold water mass, but only verified it on the multi-year average seasonal cycle (Figure 2b), and concluded that "Therefore, this model is suitable for examining the factors influencing the interannual variation in

the INCWM via sensitivity experiments. "More comparisons between obs. and model results are needed to confirm model validation, including episodic and interannual time scales, transections comparisons.

Thanks for your comments. In the study, the model is driven by multi-year averaged daily surface fluxes of momentum, heat and fresh water. The primary aim of using numerical model is to study the response of INCWM to atmospheric changes by sensitivity experiments. Under the circumstances, a climatological model is applicable to do some sensitivity numerical experiments. Therefore, we validated the seasonal evolution of INCWM by the comparison with multi-year averaged observation data. We add information about the model configuration in Section 2.2 to make it clear.

4. Does the CONTROL run only have seasonal climatology or actually it is an interannual hindcast of this region? It should be clearly stated in the main text. If it only has seasonal climatology, it is not reasonable to compare with the three sensitivity runs, and evaluate the contributions of air-sea heat flux. If it is actually a hindcast, why authors do not show any interannual comparisons with observations to validate the model? Without validation of this model on interannual time scale, it's really hard to trust and interpret the interannual results from the sensitivity runs. On the other hand, if authors do have a hindcast of this model from 1994-2015 and it shows solid results comparing with observations, it is pretty interesting to quantify the cold water mass properties on interannual time scale using the model as well. Because the observational dataset is only present at 8 stations with very coarse spatial resolution, however, the model has a three dimensional distribution of the cold water mass.

Thanks for your comments. we used climatological forcing (multi-year average) in numerical model rather than an interannual hindcast. We add information in Section 2.2 to make it clear. In the study, we investigated the interannual variation of INCWM using long-term observation for the first time. The model was used for just evaluating the response of INCWM properties to sea surface forcing changes. As you said, we plan to conduct a hindcast to quantify the INCWM properties on interannual time scale detailly as a continuation of this work. Your comments also give us a lot of help. We show the shortcoming of the work and future plans in the last paragraph in Section 5.

For the three-dimensional distribution of INCWM, we add the horizontal and vertical

distribution of simulated INCWM based on model results in Section 2.1 and Section 4.1.

5. Ln 136-141: How the f time series is calculated? Is it area-averaged for each domain? How about delta f? Do they have units? If so, please add them to Table 2. The q is the absolute value of the relative change of Temp/Area. Therefore, it is not sure if it is a decrease or an increase in Temp/Area in sensitivity runs. Also, in Table 2, the q value for each case is ground to one value. Is it a multi-year average? More details needed for the calculations.

Thanks for your comments. The calculation of q is changed as  $q = \frac{T_{case} - T_{control}}{T_{control}}$  to show a decrease or an increase in Temp/Area in sensitivity runs. The q is calculated to demonstrate the relative change of INCWM characteristics between two cases (sensitivity run and Control). It is not a multi-year average. We changed the description in Section 4.1. As answered above, the content about cr is removed in the revised manuscript. As for f, it is only used in the calculation of cr. Therefore, we deleted the content about index (cr) and f in the revised manuscript.

6. Ln 193-194: April is selected as the initial month of the water mass formation, but figures only starts from April. To make the point, authors are suggested to show correlation panels at least for March as well. Ln95-97: the authors stated that "water temperature in April depended mainly on the cooling process, the initial temperature of the INCWM is likely associated with local air-sea heat flux from winter to early spring", any reference for it?

Thanks for your suggestion and comment. As your suggestion, we added the figures of correlation coefficients from previous December to March in Fig. 5 which have little significant correlation.

The sentence "water temperature in April depended mainly on the cooling process, the initial temperature of the INCWM is likely associated with local air-sea heat flux from winter to early spring" locates at Ln 195-197. Before April, the water mixes well under surface cooling and increasing wind in the previous winter, the air-sea sea heat controls the process of surface cooling, therefore, the air-sea heat flux is closely related to the water temperature at the early spring. Tsutsumi and Guo (2016) suggested the heat content inside

the Seto Inland Sea (except for Bungo Channel and Kii Channel) from January to July mainly depended on air-sea heat fluxes through heat budget. Therefore, air-sea heat flux is an important factor for water temperature in April.

We changed the sentence as "The water temperature in April is the result of cooling process in the previous winter. Tsutsumi and Guo (2016) suggested the heat content inside the Seto Inland Sea (except for Bungo Channel and Kii Channel) from January to July mainly depended on air-sea heat fluxes. the initial temperature of the INCWM is likely associated with local air-sea heat flux from winter to early spring".

**Reference:**

Tsutsumi E, Guo X. Climatology and linear trends of seasonal water temperature and heat budget in a semi-enclosed sea connected to the Kuroshio region. Journal of Geophysical Research: Oceans, 2016, 121(7): 4649-4669.

 In 198-199: "correlation coefficient below 10 m is larger in May-July than in April (Figs. 4b-d)" Actually it is not true based on Figure 4. Most of the markers show almost the same values, some even decreases, only two particular markers show the increase.

Thanks for your carefully reading. We have changed the colormap to make the information clear. From the Figure 5 in the revised manuscript, it is clear to show that the correlation coefficient below 10 m in Iyo-Nada (Sta. 1-7) increased from April to May-July, especially May and July. In addition, the significant difference levels gradually enhanced from 0.01 to <math>p < 0.01 from April to July.

Correspondingly, we changed the sentence as "As Fig. 5e-h shown, the correlation coefficient below 10 m in Iyo-Nada (Sta. 1-7) increased from April to May-July, especially in May and July. In addition, the significant difference levels gradually enhanced from 0.01 to <math>p < 0.01 during this period, which indicated that heat transport into the INCWM from May to July is also important for the interannual variation of the INCWM in July." in the second paragraph in Section 3.3.

Figure 5. The blue dots does not match those in Figure 3. Some dots are not denoted with years.
Please add their years in the figure.

Thanks for your careful reading. We checked the dots and corresponding years and found

two strong years (2006 and 2008) were omitted in Fig. 4. As your suggestion, we added table (Supplementary Table 1) to show strong and weak years of INCWM and replot Fig. 4 and Fig. 7 in the revised manuscript.

9. Ln 245-247: This is a really confusing conclusion. cr(A) and cr(T) provide opposite results, which is obviously weird, as T and A are always strongly related, but clearly authors choose to trust cr(T) instead of cr(A). More explanations needed here.

As mentioned above, according to peer reviews and our deep thinking, *cr* has no significant physical significance and is not suitable to account for the contribution of one factor on interannual variation of INCWM. Therefore, the content about *cr* is deleted in the revised manuscript.

Ln 253-256: Winds are weak during the warming season, how about cooling season, Dec-Feb? Case
1 should not be small, right? 0.006 and 0.06 are for case 2 and 3?

Sorry for the rough description caused you to misunderstand. We mean that all the sensitivity coefficient in the experiments about wind are less than 0.00r and 0.06 for the temperature and area of the INCWM in the old manuscript. In the revised manuscript, considering the presence of essential strong wind in summer in the climatological model, we removed the results of wind experiments, and focus on the results of experiments about air-sea heat flux.

Section 4.2 Repeated texts for the different cold water masses. e.g. Ln 275-279 vs. Ln 323-329
Thanks for your comment. We removed description on Ln 323-329.

**Minor Comments:**

 Some sentences need to be improved in language., e.g. Ln 49-50: This sentence is suggested to be rephrased: "This study focuses on ...".

Thanks for your suggestion. We rephrased the sentence as "*In this study, we focuses on the BCWM in Iyo-Nada, which connects the Bungo Channel with the Hayasui Strait (Fig. 1b)*" in the third paragraph in Section 1.

13. Ln 148: 1994-2012 or 1994-2015?

In the old manuscript, the air-sea heat flux and wind stress provided by Japanese 55 year

Reanalysis (JRA55) are from 1994 to 2012. In the revised manuscript, we removed the relevant content about the index cr, so we deleted the information about the JRA 55 dataset which only used in the calculation of cr.

14. Ln 169: Figure 3 and caption do not match. It says authors used "18.23°C" as the average temperature, but the figure suggests 18°C for that year. Should it change to "18°C" to match the figure?

Thanks for your comment. In the old manuscript, we used a red star to indicate 18.23°C but plot at the location of 18°C for facility performance. In the revised manuscript, we changed the red star in Fig. 4 to match 18.23°C.

15. The color scheme of Figure 4 is too similar from 0.45 to 0.75, and the markers are too small. Values where p>0.05 are also suggested to show, in order to see the correlation patterns.

Thanks for your suggestion. We have changed the colormap and size of markers in Figure 5, and show the correlation coefficients with p>0.05 to see the correlation patterns.

Figure 5: Correlation coefficients for interannual variation of water temperature at each station from previous December to July (a-h) and that of the Iyo-Nada bottom cold water mass (INCWM) mean water temperature in July during 1994-2015. Circles indicate that the significant difference level is less than 0.95, squares indicate a significant difference level is between 0.95 and 0.99, and stars indicate a significant difference level of more than 0.99.

---

## Author Comment (AC3)

Referee comment on "Air-sea heat flux during warming season determines the interannual variation of bottom cold water mass in a semi-enclosed bay" by Junying Zhu et al., Ocean Sci. Discuss., https://doi.org/10.5194/os-2021-96-RC3, 2021

**General comments**

In this study, the determining factors of the intensity of cold bottom pools in shelf seas and bays are studied. The major study site is a semi-enclosed bay in Japan, but also generalisations to comparable bay are made. The major conclusion is that the strength of the cool pool sometimes depends on the previous winter SST (other studies) or on the air sea buoyancy flux during the warming season (this study). While these results are potentially interesting, I think that their value is limited here, since the methods used are not state of the art. The quantification of the cold pool strength depends on a highly site-specific empirical measure, rather than on energy considerations.

Furthermore, the applied numerical model uses climatological forcing rather than realistic forcing. With this, a comparison between model results and field observations is not possible and interannual variability, a major focus of this study, cannot be assessed. For these reasons, I recommend to reject the manuscript at this stage and motivate resubmission of a manuscript that uses state-of-the-art methods.

Thanks for your careful reading and comments. Following your comments, we have finished a comprehensive revision on the original manuscript. Below is a point-to-point response.

In this study, we demonstrated the interannual variation of INCWM and explored its influencing factor using observation and numerical model. For the bottom cold water in Seto Inland Sea (Japan), this paper is the first time to report its interannual variation, therefore, we focused on its interannual variation and preliminarily analyzed influencing processes.

As you said, the quantification of the cold pool strength depends on a highly site-specific empirical measure. The prefectural fishery research centers carried out regular hydrographic observations at monthly intervals in the Seto Inland Sea (stations are shown in Fig. 1). Yu et al. (2016) studied the location change of front around INCWM using the data from Station 1 to Station 7 (Fig. 1 here). They also showed the temperature difference between the surface layer (0 m) and bottom layer (the deepest sampling depth at each station) in the climatology data from April to September (Fig. 2 here). As Fig. 2 (below)shown, INCWM occurs from May to September and locates at the central area of Iyo-Nada. The temperature difference between surface and bottom reaches its maximum in July. The transection (Sta.1 to Sta. 7 in Fig. 1) we used in the study just across the middle of INCWM. Therefore, the measurements of water temperature along the transect from Sta. 1 to Sta. 7 are supposed to be suitable to explore its cold pool strength and interannual variation. In our model, the location of INCWM (Fig.3 in the revised manuscript) is consistent with observation.

In the development of cold pool when water volume changes from mixed water to thermocline, heat transport process is the key to maintain INCWM and influence its strength. Therefore, based on the results of observation, we considered the energy change when analyzing the influencing factor of

INCWM strength. In the discussion part, we compare the different BCWMs in different coastal seas from the viewpoint of energy. We thought that highly site-specific empirical measure and energy considerations are complementary. The discussion about energy helps us to understand BCWMs variation with limited observation. Of course, more site-specific empirical measure is indispensable for a deeper understanding of BCWMs.

In terms of numerical model, we actually use climatological forcing in numerical model rather than realistic forcing. The primary aim of using numerical model is to study the response of INCWM to atmospheric changes by sensitivity experiments. Under the circumstances, a climatological model is applicable. Therefore, we ran model using climatological forcing and validated model using climatology observation data. This is our first work to study the interannual variation of INCWM, and in the future work, we will apply realistic forcing to drive model for intensive study on interannual variation of BCWMs in the Seto Inland Sea.

[Figure]

Fig. 1 All observation sites in Seto Inland Sea. Sta. 1-7 is used to study INCWM in our study.

[Figure]

Fig. 2 Temperature difference between the surface layer (0 m) and bottom layer (the deepest sampling depth at each station) in the climatology data from April to September, obtained as averages of

data collected in the same months between 1971 to 2000. (from Yu et al., 2016)

Note: We collected the observation data from 1971 to 2015. However, the observation of Sta. 4 (around the central area of INCWM) starts from 1994, so we analyzed the interannual variation of INCWM from 1994 rather than 1971.

**Specific comments**

32-34: What defines a BCWM to be strong or weak? The temperature of it will certainly increase during spring and summer, such you probably define strength thought some temperature differences? Please specify.

Thanks for your suggestions. We have already considered to define BCWM intensity using temperature difference between spring and summer. However, this definition ignores the retention of water temperature in winter and only considers the temperature change from spring to summer. it is not suitable to define the NCWM intensity. Since BCWM steadily occurs almost every year, its performance characteristics, such as temperature and area, changes every year, are appropriate indicators to indicate its intensity. Therefore, we used the characteristics of water temperature and area of INCWM to define its intensity.

37: better "… and the Middle Atlantic Bight Cold …"

Thank and changed as your suggestion.

40: Isn't it more simply and more directly the winter temperatures of the vertically well mixed shelf sea waters and then probably also the summer SST that determine the strength of the BCWM? In addition to the heat fluxes, also laterally advective exchanges could set the temperatures. This is basically what you argue in lines 41-43.

Thanks for your comment. In lines 41-43, we just presented the result of Chen and Curchitser (2020) about the Middle Atlantic Bight Cold Pool whose interannual variation is related to the previous winter temperature and abnormal warming/cooling due to oceanic advection (including vertical and horizontal advection). In the paper, they also concluded that the winter (mid-January to March) temperature anomaly was the primary factor in determining the interannual variability of temperature anomaly near bottom cold pool region during the stratified seasons by a long-term numerical simulation. We did not emphasize the influence of laterally advective exchanges.

We are sorry that the improper expression caused your misunderstanding, we changed the sentence as "*Chen and Curchitser (2020) suggested that its temperature interannual variations during stratified seasons were controlled by both the previous winter temperature and abnormal warming/cooling due to total oceanic advection, and the winter (mid-January to March) temperature anomaly was the primary factor in determining the interannual variability of temperature anomaly near bottom cold pool region during the stratified seasons*." in lines 45-49 in

the revised manuscript.

In terms of the factors determine the strength of the INCWM in this study, we calculated the correlation coefficients between the average temperature of the INCWM in July and the water temperatures at all the depths of each station from 1994 to 2015 for each month (from previous December to July). As shown in Fig. 5 in the revised manuscript, the water temperature of INCWM in July is not significantly related to winter well-mixed water temperatures and so is the summer SST. So, we could not use them to determine the strength of the INCWM. This also demonstrated the difference in control factors between INCWM and other bottom cold water masses (the Yellow Sea Cold Water Mass and the Middle Atlantic Bight Cold Pool).

74: make sure that you avoid double brackets ")(".

Thank and we have rewritten the sentence to avoid double brackets in the first paragraph in Section 2.1.

94-96: This measure certainly depends on the position of the transect relative to the cold water pool. Therefore, it is not a suitable measure. Also temperature itself is not a good measure, because it is too site-specific. Temperatures differences between surface and bottom need to be involved. A better measure would for example be the thermal contribution of the potential energy anomaly, integrated over an entire bay. Or you could use the thermal contribution to the Available Potential Energy (APE) of the bay. This can easily be calculated by means of a numerical model. Measurements can be used to reconstruct this as well, when some assumptions about the geometry of the cold pool are made. The measure could be converted into a mixing time scale by division by the kinetic energy supply through tides and wind (plus/minus surface buoyancy flux contributions).

Thanks for your comments. As mentioned before, the measure in this study across the central area of INCWM. Although there is only a vertical transection, it is the best measured data for analyzing the long-term variation of INCWM at present. We add description about the seasonal temperature difference between surface and bottom in the revised manuscript (Fig. 3).

This is the first time to explore the interannual variation of INCWM using long-term regular observation data. In this study, we focus on showing the interannual variation of INCWM and preliminary discuss its influence factor. As you said, more stations and observed hydrological parameters are needed to clarify detail dynamic mechanism controlling interannual variation of INCWM. On the basis of this work, we plan to perform more detailed observations and model simulations in the future to deeply study the dynamic mechanism, your suggestions give us many helps.

96: With the two indices you probably mean the transect area and the temperature.

The two indices are the averaged water temperature inside the INCWM and the area of INCWM

along the observational transect. For better reading, we changed the sentence as "the intensity of the INCWM was defined by two indices, i.e., the spatially averaged water temperature inside the INCWM and the area of INCWM along the observational transect.". And move the description from Section 2.1 to Section 3.1.

100: This measure is highly empirical and not physically based, see above.

Kindly, the prefectural fishery research centers carried out regular hydrographic observations at monthly intervals in the Seto Inland Sea (stations are shown in Fig. 1) to monitor hydrological conditions and fishery resources. The stations are designed based on both physical oceanography and fishery distribution. High primary production and abundant fishery resources usually occurs where the cold water mass locates in many coastal seas (Yoon et al., 2000; Narváez et al., 2015; Abe et al., 2015; Coakley et al., 2016). As shown in Fig. 1 and Fig. 2 in the revised manuscript, the long-term observation transection (from Sta. 1 to Sta. 7) across the interior of INCWM and could present the temporal and spatial variation of INCWM well. The observation data is the best field survey data we can find to analyze the variation of INCWM although it just shows the vertical structure of INCWM.

References:

Abe K, Tsujino M, Nakagawa N, et al. Characteristic of Si: P: N ratio in bottom water in central Suo-Nada, western Seto Inland Sea. Journal of oceanography, 2015, 71(1): 53-63.

Coakley S J, Miles T, Kohut J, et al. Interannual variability and trends in the Middle Atlantic Bight cold pool//OCEANS 2016 MTS/IEEE Monterey. IEEE, 2016: 1-6.

Narváez D A, Munroe D M, Hofmann E E, et al. Long-term dynamics in Atlantic surfclam (Spisula solidissima) populations: the role of bottom water temperature. Journal of Marine Systems, 2015, 141: 136-148.

Yoon W D, Cho S H, Lim D, et al. Spatial distribution of Euphausia pacifica (Euphausiacea: crustacea) in the Yellow Sea. Journal of Plankton Research, 2000, 22(5): 939-949.

115-117: The model is forced by some kind of climatological wind, which leads to underestimation of the wind-energy input and mixing. The method for calculating the surface buoyancy flux is not mentioned. Since the wind is climatological, the reviewer can assume that also the buoyancy fluxes are idealised. Also, no information is given about open boundary conditions and riverine freshwater forcing. For an investigations like this one, is would be state of the art to apply a model with realistic forcing. Some information on the surface buoyancy forcing is given in lines 146-151, and it seems indeed that this forcing is climatological as well.

The following sections include some interesting discussions, but since the study is based on a highly site-specific empirical measure for the size of the cold pool and the numerical model is highly idealised, I propose that the authors do first improve their methods according to the above suggestions and then repeat the study.

Thank for your comment.

We actually use climatological forcing in numerical model rather than realistic forcing. The detailed model configurations have added in the first paragraph in Section 2.2.

"*Four major tidal constituents (M2, S2, O1 and K1) at the open boundary were considered and the daily river discharges averaged over 24 years (1993–2016) from the Ministry of Land, Infrastructure and Transport were used in the model. Multi-year averaged daily surface fluxes of momentum, heat and fresh water was used to drive model (Zhu et al., 2019). The daily wind stress was based on hourly averaged results of wind stress, which was calculated by wind velocity from the Grid Point Value of Meso-Scale Model (GPV-MSM) (http:// database.rish.kyoto-u.ac.jp/arch/jmadata/data/gpv/) during 2007–2016 provided by the Japan Meteorological Agency with the resolution of 1/16°×1/20°, adopting the drag coefficient of Large and Pond (1981). The daily shortwave radiation was based on the newly released of Japanese Ocean Flux Data Sets with Use of Remote Sensing Observation (J-OFURO3) (https://j-ofuro.scc.u-tokai.ac.jp/) with a resolution of 1/4°×1/4° and averaged during 2002 to 2013. The daily longwave, sensible heat flux and latent heat flux were calculated and averaged by adopting bulk formula (Gill, 1982) using hourly air temperature, sea surface temperature, relative humidity, cloud cover, and wind velocity from the GPV-MSM (2007–2016). Daily evaporation was obtained by calculating the latent heat flux. The daily precipitation was provided by the GPV-MSM and averaged hourly from 2007 to 2016*" The seasonal variation is shown in below figure.

The information in lines 146-151 in the old manuscript is about the dataset JRA55 which has been removed in the new manuscript.

Kindly, we first demonstrated the interannual variation of INCWM and explored its influencing factor in the study using long-term observation and a numerical model.

We are the first time to explore the interannual variation of INCWM and its influence factors. Model driving by climatological forcing could capture the evolution of INCWM and it can be competent to discuss the response of INCWM to air-sea heat flux changes by sensitivity numerical experiments. Based on this work, we plan to conduct a continuous run with realist forcing to detailly explore dynamic mechanism controlling the interannual variation of INCWM. We show the shortcoming of the work and future plans in the last paragraph in Section 5.

[Figure]

Fig. 3. Daily variations in multi-year averaged (a) wind speed, (b) wind direction, (c) air-sea heat flux, and (d) river discharges used in case Control. The wind direction in (b) is clockwise and the direction of southerly wind is 0. Positive values in (c) indicate that the ocean gains heat, while negative values indicate the opposite situation. (from Zhu et al., 2019)

---

## Referee Report (RR1)

Review of OS-2021-96

The manuscript has been improved, and I am satisfied with the way how the authors address the reviewers' comments. I only have these following questions/comments:

1. Sensitivity experiments: when comparing Case 2 and Case 3, actually it is unfair to compare their impacts; Case 2 increases surface heat gain locally by 10% from May to July, which lasts for 3-months; while Case 3 increases surface heat gain in Hayasui Strait by the same percent, but only for one month - July. And the two cases' impacts are measured and compared for July. It is reasonable to think, the less temperature change in July for Case 3 than that for Case 2 is because the heat-flux-increase's lasting time period is shorter. To avoid this problem, Case 3 or a new case should be modified or added: increase surface heat gain in Hayasui Strait by 10% from May to July.

2. Lines 470-471. "This is an inverse pathway to the heat transport and is expected to be large in some BCWMs. " What does this sentence mean?Why nutrient transport is an inverse pathway to heat transport?

Minor:

1. Line 58: "… a schematic diagram in Fig. 5" - >  Fig. 6 ?
2. Lines 90-91: "However, water temperature at Hayasui Strait is almost **vertically** homogeneous throughout a year"
3. Line 105: "The red line of the 18 °C isotherm in (a)" -> (g)
4. Line 106: "The red and black bars in (b) " -> (m)
5. Line 129: analysis -> analyze
6. Lines 173 and 299: "Fig. 2c" -> Fig. 2g
7. Line 178: "Supplementary Table 1" -> 2
8. Supplemental Figure 4 caption: July -> August
9. Figure 5 caption: significant difference level? -> significant confidence level; as well as in Line 249
10. Line 233: Fig. 5a -> Fig. 5e; Figs. 5b-d -> Figs. 5f-h
11. Line 236: Fig. 5d -> Fig. 5h

---

## Author Response (AR2)

**Response to Reviewer**

**Comments to the author**:

Dear Dr. Guo,

Thank you for your thorough revision of the previous version of the manuscript. The reviewers have returned only minor comments and seem satisfied with your revision. Before we take the manuscript further, however, I must request some changes.

This is not a climate-change effect study. If the intention was so, the entire study must be redesigned and is not acceptable in its present form. I would strongly recommend you carefully go through the manuscript and edit any part relating to climate change effects. Some statements can be retained as speculative comments, but please tone down in the discussion, abstract, results and conclusions. The final statement of the abstract "These findings will help to predict bottom water temperatures and improve the current understanding of ecosystem changes in shelf seas under global climate change" must be removed or revised as I cannot see how your results will improve the current understanding of ecosystem changes in shelf seas.

A careful reading/edit is necessary. In some parts, the language is not clear. Below are some line-by-line comments, not separated minor/major.

Thanks for your careful reading and comments. This study investigates the interannual variation of INCWM and its response to air-sea heat flux change. We have carefully gone through the manuscript and edited the part relating to climate change effects. The final statement of the abstract changed to "These findings will help us to understand the response of bottom cold water mass in coastal seas to sea surface forcing change". Following your comments and suggestions, we have finished a comprehensive revision on the original manuscript. Below is a point-to-point response.

Li14: [its] influencing factors- clarify ambiguity

Thanks for your comment. We changed the statement to "its response to air-sea heat flux change".

Li 15: [t]he lower case

Changed as your suggestion.

Li16: [Their] internannual variation- clarify ambiguity

Thanks for your comment. We changed the statement to "The interannual variation of average water temperature of INCWM"

Li36: dissipating: wrong word (water mass does not dissipate). You mean disappearing?

Thanks for your careful reading. We changed the word to "disappearing".

Li46: [T]hey, upper case

Thanks and changed as your suggestion.

Li 51: Kii Channel is not marked in fig 1

Thanks and we have marked the "Kii Channel" in new Fig. 1a.

Li 53-54: only one of these place names are marked in fig 1

Thanks for your comment. We have marked the location of Suo-Nada, Iyo-Nada, Hiuchi-Nada and Harima-Nada in new Fig. 1a.

Li 54: we focus[es]: delete "es"

We have deleted "es" as your suggestion.

Li 64: "changes under climate change": this motivation is good but not addressed thoroughly in your study. Please consider removing altogether.

Thanks for your comment. We have removed "under climate change".

Li81: no need to give the URL for a research institute.

We have deleted the URL as your suggestion.

Li83: use oC, degree

We have changed the word "degree" to "°C".

Li 88-100: This part is not method. It belongs to a subsection in results (e.g., call it Seasonal variability)

Thanks for your comment. We have moved this part to a new section "Section 3.1 Seasonal variability of the INCWM".

Li 95-100, and thoughout, "deep" must be "depth" when referring to measurements from XX m depth.

Thanks for your comment. We have changed "deep" to "depth" when referring to measurements from XX m depth in the revised manuscript.

Colormap: please use a better-suited colormap than "rainbow/jet" for temperature in Fig 2 & 3. For Fig 9 consider a blue-white-red colormap. One suggestion is to use the cmocean package available for Matlab and other common softwares.

Thanks for your suggestion. We have changed the colormap in Fig.2 and Fig. 3 using the cmocean package available for Matlab. For Fig.9, we consider that you said is Fig. 8, and we have replotted with a blue-white-red colormap.

Fig 2 caption: bars in (m) not (b). 50 m depth.

Thanks and changed as your suggestion.

Li 118: were used to force the model

Thanks and changed as your suggestion.

Li140: delete obvious

We have deleted "obvious".

Li 132 to 150: these are also "results", not methods. Please restructure.

Thanks for your comment. We have moved this part to a new section "Section 3.1 Seasonal variability of the INCWM" which compare the observed and simulated seasonal variability of the INCWM. Meanwhile, we changed the title of Section 2.2 to "Model configuration".

Li 173: choice of 18C isotherm: seems arbitrary. Please discuss the sensitivity to this choice.

Thanks for your comment. We choose 18 ℃ isotherm based on properties of the cold water mass. First, a bottom cold water mass always shows a shape of bottom-up "bowl" that encloses a water mass there. The observed distributions of temperature (Supplementary Fig. 3) show that the water less than 18 ℃ always occupies the central area of Iyo-Nada below 20 m in all observed years and the form of 18 ℃ isotherm is similar to a bottom-up "bowl". Second, the water temperature difference between surface and bottom layers is an important index to identify the INCWM. From May to July (Fig. 3b-d), the water temperature difference between surface and bottom layers increases faster in the area defined by 18 ℃ isotherm for the INCWM than in the surrounding waters. A bottom temperature front gradually separates the INCWM from the surrounding vertically mixed water. In July, INCWM can be well distinguished from surrounding waters by a water temperature difference between surface and bottom layers of 5 ℃ (Fig. 3d), whose location is consistent with that of 18 ℃ isotherm at 50 m depth (Fig. 3g). Therefore, the use of 18℃ isotherm is not arbitrary. To clarify this point, we added following

sentences in the first paragraph in Section 3.2.

"Observation (Supplementary Fig. 3) shows that INCWM occupies the central area of Iyo-Nada below 20 m depth with a similar shape of bottom-up "bowl" which can be indicated by the 18 ℃ isotherm in almost all years. In addition, INCWM in July can be well distinguished from surrounding waters by the water temperature difference between surface and bottom layers of 5 ℃ (Fig. 3d) whose location is consistent with that of 18 ℃ isotherm at 50 m depth (Fig. 3g). Therefore, we used the 18 ℃ isotherm to define the boundary of INCWM in July (Fig. 2c), and calculated the average water temperature and area inside the 18 ℃ isotherm."

Li 219-220: consider: "..temperature at the INCWM site was homogeneous….dome below a surface layer warmed by surface heating…."

Thanks and changed as your suggestion.

Li253: specify what you mean by stronger INCWM (colder?)

As defined by Section 3.2, a stronger INCWM has a lower water temperature and a larger area. Here, a significant negative correlation (r = -0.44, p < 0.05) was obtained for the thermocline strength in July and the average water temperature of the INCWM in July, indicating that a stronger thermocline strength corresponds to a colder INCWM. To be clear, we changed "a stronger INCWM" to "a colder INCWM".

Li263: not all of these are "heat transfer" processes. Without loss of information, you can delete and use "by three processes"…

Thanks and we have deleted "heat transfer" in the sentence.

Li 272: However, the local vertical heat… [delete "it is noted to say that"]

Thanks and changed as your suggestion.

Fig 6. What is "gravitational circulation"? Describe or simply call it mean circulation or similar.

In summer, the increase of river discharge into the Seto Inland Sea induces a density circulation which occurs as the bottom water flows from the Hayasui Strait to Iyo-Nada while the surface water flows in the opposite direction. The circulation is similar with estuarine gravitational circulation and we draw this circulation in Fig. 6. To be more accurate, we changed "gravitational circulation" to "density-driven circulation" in Fig. 6b.

Li295: consider: "q, defined as the relative ….is used to quantify the response"

Thanks and changed as your suggestion.

Li305: surround[ed] by [the] 18oC isotherm

Thanks and changed as your suggestion.

Li310: delete "obviously"

Thanks and we have deleted "obviously".

Li368: response to climate change is not studied.

We have changed "climate change" to "sea surface forcing change".

Fig 9b: the gray arrows. I do not understand how you can link different ocean regions like this (and bypass the Irish Sea). It is confusing and must be removed.

Thanks and we have removed the gray arrows in Fig. 9(b).

Li449.451: this summary of mean temperatures etc. is not a conclusion

Thanks for your comment. We have deleted the summary of mean temperature and added the content about the intensity of INCWM as a part of conclusion.

"The INCWM was strong in the years of 1994, 1996, 2006, 2010, 2012, 2013, and 2015, while weak in the years of 1997, 2003, 2004, 2007, 2009, 2011, and 2014."

Li466: I disagree that the used a "climatological model"

Thanks and we have changed to "a hydrodynamic model".

---

## Author Response (AR3)

**Response to editor**

**Comments to the author**:
Thank you for addressing my comments.

I cannot see your response to referees #1 and #3. They also made minor comments you should consider and respond.

When preparing the next version also note the comment from Polina Shvedko (21 March).

Additionnaly, please consider using cmocean's "thermal" colormap for the temperature fields of Fig 2 and 3, and a blue-white-red (0 centered at white) for Figure 5 (correlation coefficients).
Thank you,

Ilker

Thanks for your kind reminder and suggestion. We have made responses to referees #1 and #3 in the new revised manuscript. Following your suggestion, we have changed the colormap to cmocean's "thermal" for the temperature fields of Fig. 2 and Fig.3 and a blue-white-red for Fig. 5.
According to the comment from Polina Shvedko, we have renamed the figures and tables in supplement file, such as "Fig. 1" to "Figure S1" and "Table 1" to "Table S1", and modified the representation in the new text when it comes to the corresponding diagrams and tables. Besides, we are the originator of the images (Figure 1, Figure 3 and Figure 8), we have confirmed this issue explicitly by email.
We express our thanks for editor and reviewers in the new revised manuscript.

**Response to review #1:**

Regarding comment #1 in my first review:
As the authors emphasize, the ideal model experiment shows that surface heating from May to July has a significant effect on the bottom cold water. On the other hand, the observation result (Fig. 4) shows that the surface temperature during this period does not correlate with the temperature of the bottom cold water. Considering that surface heating first changes the surface temperature, there seems to be a contradiction between the model and observation results. The authors should discuss possible causes of this contradiction, or note that it should be resolved in the future.

Caption of supplementary Fig. 4: Correct "July" to "August".

Thanks for your careful reading and comments.

The water temperature of bottom cold water mass is influenced by vertical heat transfer from upper layer water and horizontal heat transport from lateral water. In summer, vertical heat transfer from sea surface is impeded by the strong thermocline. The vertical distribution of temperature difference between Case 2 and Control (Case 2 - Control) from May to July is shown (Fig. R1). When local sea surface heating increases by 10% in Case 2, the temperature difference at surface ranges from 0.2 ℃ to 0.5 ℃, while the temperature difference inside the INCWM is around 0.05 ℃ - 0.1 ℃. The 0.1 ℃ isotherm of temperature difference always appears at around the depth of 10-20 m where the thermocline develops from May to July (Fig. 2e-g). Apparently, more heat stays above the thermocline while the INCWM has the same variation of water temperature with the waters around the depth of 10-20 m. The increase of INCWM temperature is obviously slower than that at sea surface when increasing local heat flux. Besides, the lateral heat transport due to the presence of density-driven circulation is an important process for temperature variation of INCWM according to sensitivity numerical experiment and observation.

In the ideal model experiment, we show that the temperature change in a BCWM caused by the interannual variation of air-sea heat flux is proportional to sea surface heat flux change under the assumption of constant vertical and horizontal heat transport velocity. In this case, the change of BCWM temperature should be consistent with that of surface temperature although the thermocline reduces the vertical heat transport. However, we do not consider the changes in density-driven current which is mainly influenced by river discharge and the water temperature in Hayasui Strait. The lateral heat transport could readjust the temperature change of the cold water mass, weakening the consistency between changes of BCWM temperature and sea surface temperature. Correlation analysis from observation (Fig. 5f-h) shows that the water temperature of INCWM in July is closely related to the water temperature around the depth of 10-20 m, but is not correlated with sea surface temperature. This is the result of above two heat transfer processes, which also illustrates the importance of lateral heat transport for temperature variation of BCWM in Iyo-Nada.

Therefore, the results of ideal model and observation is not contradiction. In Fig. 9, we only evaluate the importance of vertical heat transport caused by change of air-sea heat flux based on ideal model. According to your comment, we added the discussion about the limitations of the ideal model in Section 4.2 and related discussion of sensitivity numerical experiment in Section 4.1 to address this issue in the new revised manuscript.

Also, we have changed the "July" to "August" in the caption of supplementary Fig. S4.

**We added the description below in Section 4.1:**

"Compared the vertical temperature distribution from May to July between Case 2 and Control (Fig. S5), when local sea surface heating increased by 10% in Case 2, the temperature difference at surface ranged from 0.2 ℃ to 0.5 ℃, while the temperature difference inside the INCWM was 0.05 ℃ - 0.1 ℃. Meanwhile, the 0.1 ℃ isotherm of temperature difference (Case 2 - Control) always appeared at around the depth of 10-

20 m where the thermocline develops from May to July (Fig. 2e-g). Apparently, more heat stayed above thermocline while the INCWM mainly obtained heat from waters around 10-20 m."

**We added the description below in Section 4.2:**
"In the evaluation, local air-sea heat flux is key factor for interannual variation of INCWM. However, correlation analysis from observation (Fig. 5f-h) shows that the water temperature of INCWM in July is closely related to the water temperature around the depth of 10-20 m, but is not correlated with sea surface temperature which responses to air-sea heat flux first. On the one hand, due to the presence of a strong thermocline in summer, the response of INCWM temperature is obviously slower than that at sea surface when local air-sea heat flux changes (Fig. S5). On the other hand, we do not consider the changes in horizontal density-driven current in the above evaluation (Fig. 9b) which is mainly influenced by river discharge and the water temperature in Hayasui Strait. The lateral heat transport could readjust the temperature change of the INCWM, weakening the consistency between changes of INCWM temperature and sea surface temperature."

[Figure]

Figure R1 Vertical distribution of temperature difference between Case 2 and Control (Case 2 - Control) from May to July.

**Response to review #3:**

The manuscript has been improved, and I am satisfied with the way how the authors address the reviewers' comments. I only have these following questions/comments:
Thanks for your affirmation and suggestions. Following your comments, we have finished a comprehensive revision on the original manuscript. Below is a point-to-point response.

1. Sensitivity experiments: when comparing Case 2 and Case 3, actually it is unfair to compare their impacts; Case 2 increases surface heat gain locally by 10% from May to July, which lasts for 3-months; while Case 3 increases surface heat gain in Hayasui Strait by the same percent, but only for one month - July. And the two cases' impacts are measured and compared for July. It is reasonable to think, the less temperature change in July for Case 3 than that for Case 2 is because the heat-flux-increase's lasting

time period is shorter. To avoid this problem, Case 3 or a new case should be modified or added: increase surface heat gain in Hayasui Strait by 10% from May to July.

Thanks for your suggestion. According to the results of observation (Section 3.3 and Section 3.4), we found that the temperature change of INCWM in July on an interannual scale is controlled by three processes, i.e., the local retention of bottom low water temperature from early spring, local vertical heat diffusion from May to July and horizontal heat advection originating from Hayasui Strait in July. Therefore, we designed three sensitivity numerical experiments to analyze the influence of air-sea heat flux during the corresponding time period.

Actually, we have did a sensitivity numerical experiment with the case that sea surface heat gain in Hayasui Strait from May to July increased by 10% (Case 4 in the new manuscript). Results show that the sensitivity coefficients have little difference with those in Case 3 which suggests the lateral heat transfer induced by air-sea heat flux change in July is the most important than those in May and June. Comparison between Case 3 and Case 4 also confirms the correlation analysis from observation (Fig. 5h) in Section 3.3. Therefore, we mainly discussed the result of Case 3 in Section 4.1.

According to your suggestion, we added a new sensitivity numerical experiment named as Case 4 and related description (in Table 1, Table 2, and Section 4.1) in the new revised manuscript.

"In Case 4, the sensitivity coefficients had little difference with those in Case 3 which suggested that the effect of remote air-sea heat flux change in July is much larger than those in May and June. Comparison between Case 3 and Case 4 confirmed the importance of lateral heat transfer from Hayasui Stratit to Iyo-Nada in July which was also displayed from observation (Fig. 5h)."

2. Lines 470-471. "This is an inverse pathway to the heat transport and is expected to be large in some BCWMs. " What does this sentence mean?Why nutrient transport is an inverse pathway to heat transport?

Thanks for your comments. Beneath the thermocline, nutrient is abundant inside the bottom cold water mass in summer. Vertical nutrient transport from bottom cold water mass to waters in the upper layer is the key factor to subsurface chlorophyll-a maximum above the bottom cold water mass. This phenomenon has been observed in several coastal seas with bottom cold water mass (Takeoka, 1993; Williams et al., 2013; Fu et al., 2018). Besides, in our study, density-driven circulation in summer from Hayasui Strait to Iyo-Nada below surface layer can carry nutrient from Hayasui Strait to Iyo-Nada and then diffuse into the upper water. The nutrient transport in was proposed by Takeoka (2002) who given a diagram (Fig. R2) for nutrient transport in the Seto Inland Sea. As shown by Fig. R2, nutrient transport is an inverse pathway to the heat transport as the bottom cold water mass is a nutrient-rich pool in summer.

To be clear, we changed the sentence to "Another issue is the biogeochemical aspects around the BCWMs. As BCWM is a nutrient-rich pool in summer, the transport of nutrient across the BCWMs has considerable effects on the phytoplankton growth around the BCWMs (Takeoka, 2002). This is an inverse pathway to the heat transport and is expected to be large in some BCWMs".

[Figure]

Figure R2 Transport routes of heat (solid line) and nutrients (broken line). (Takeoka, 2002)

References:

Takeoka H. Progress in Seto Inland sea research[J]. Journal of Oceanography, 2002, 58(1): 93-107.

Williams C, Sharples J, Green M, et al. The maintenance of the subsurface chlorophyll maximum in the stratified western Irish Sea[J]. Limnology and Oceanography: Fluids and Environments, 2013, 3(1): 61-73.

Fu M, Sun P, Wang Z, et al. Structure, characteristics and possible formation mechanisms of the subsurface chlorophyll maximum in the Yellow Sea Cold Water Mass[J]. Continental Shelf Research, 2018, 165: 93-105.

**Minor:**

1. Line 58: "… a schematic diagram in Fig. 5" - > Fig. 6 ?
   We have changed it.

2. Lines 90-91: "However, water temperature at Hayasui Strait is almost **vertically** homogeneous throughout a year"
   Thanks and changed as your suggestion.

3. Line 105: "The red line of the 18 °C isotherm in (a)" -> (g)
   We have changed it.

4. Line 106: "The red and black bars in (b) " -> (m)
   We have changed it.

5. Line 129: analysis -> analyze
   Thanks and changed as your suggestion.

6. Lines 173 and 299: "Fig. 2c" -> Fig. 2g
   We have changed it.

7. Line 178: "Supplementary Table 1" -> 2
   We have changed it.

8. Supplemental Figure 4 caption: July -> August
   Thank you and we have changed it.

9. Figure 5 caption: significant difference level? -> significant confidence level; as well as in Line 249
   Thank you and we have changed it.

10. Line 233: Fig. 5a -> Fig. 5e; Figs. 5b-d -> Figs. 5f-h

We have changed it.

11. Line 236: Fig. 5d -> Fig. 5h

We have changed it.

Thanks for your careful reading. We have changed the sentences you mentioned above and examined the article carefully.